# A Unified Theory of Supervised Online Learnability

**Vinod Raman**[*]                                                            VKRAMAN@UMICH.EDU
*University of Michigan*

**Unique Subedi**[*]                                                            SUBEDI@UMICH.EDU
*University of Michigan*

**Ambuj Tewari**                                                              TEWARIA@UMICH.EDU
*University of Michigan*

**Editors:** Gautam Kamath and Po-Ling Loh

## Abstract

We study the online learnability of hypothesis classes with respect to arbitrary, but bounded loss functions. No characterization of online learnability is known at this level of generality. In this paper, we close this gap by showing that existing techniques can be used to characterize any online learning problem with a bounded loss function. Along the way, we give a new scale-sensitive combinatorial dimension, named the Sequential Minimax dimension, that generalizes all existing dimensions in online learning theory and provides upper and lower bounds on the minimax value.

**Keywords:** Online Learning, Supervised Learning, Learnability

## 1. Introduction

In the supervised online learning model, a learner plays a repeated game against an adversary over $T \in \mathbb{N}$ rounds. In each round $t \in [T]$, an adversary picks a labeled example $(x_t, y_t) \in \mathcal{X} \times \mathcal{Y}$ and reveals $x_t$ to the learner. The learner observes $x_t$, picks a probability measure $\mu_t$ over the prediction space $\mathcal{Z}$, and then makes a randomized prediction $z_t \sim \mu_t$. Finally, the adversary reveals the true label $y_t$ and the learner suffers the loss $\ell(y_t, z_t)$, where $\ell : \mathcal{Y} \times \mathcal{Z} \to \mathbb{R}_{\geq 0}$ is some pre-specified, bounded loss function. For a hypothesis class $\mathcal{H} \subseteq \mathcal{Z}^{\mathcal{X}}$ known apriori to the learner, the goal of the learner is to make predictions such that its expected regret, defined as the difference between the expected cumulative loss of the learner's predictions and that of the best-fixed hypothesis in $\mathcal{H}$, is small. We say that a tuple $(\mathcal{X}, \mathcal{Y}, \mathcal{Z}, \mathcal{H}, \ell)$ is *online learnable* if there exists an online learner such that its expected regret is a sublinear function of $T$, for any strategy of the adversary.

Given a tuple $(\mathcal{X}, \mathcal{Y}, \mathcal{Z}, \mathcal{H}, \ell)$ one is often interested in answering the following question:

> What are necessary and sufficient conditions for $(\mathcal{X}, \mathcal{Y}, \mathcal{Z}, \mathcal{H}, \ell)$ to be online learnable?

For instance, when $\mathcal{Z} = \mathcal{Y}$ and $\ell(y, z) = \mathbb{1}\{y \neq z\}$, online learnability has been characterized in terms of the Littlestone dimension of $\mathcal{H} \subseteq \mathcal{Z}^{\mathcal{X}}$, henceforth denoted as $\mathrm{L}(\mathcal{H})$. That is, $\mathcal{H} \subseteq \mathcal{Z}^{\mathcal{X}}$ is online learnable if and only if $\mathrm{L}(\mathcal{H}) < \infty$ (Littlestone, 1987; Daniely et al., 2011; Hanneke et al., 2023). Similarly, when $\mathcal{Z} = \mathcal{Y} = [-1, 1]$ and $\ell(y, z) = |y - z|$, the *sequential* fat-shattering dimension of $\mathcal{H} \subseteq \mathcal{Z}^{\mathcal{X}}$, denoted $\mathrm{sfat}_\gamma(\mathcal{H})$, characterizes the online learnability of $\mathcal{H}$. A class $\mathcal{H} \subseteq \mathcal{Z}^{\mathcal{X}}$ is online learnable if and only if $\mathrm{sfat}_\gamma(\mathcal{H}) < \infty$ at every scale $\gamma > 0$ (Rakhlin et al., 2015a). Analogous dimensions for ranking and list learning have also been established and shown

---

[*] Equal contribution

to characterize online learnability in their respective settings (Raman et al., 2024a; Moran et al., 2023).

Existing characterizations of online learnability follow three steps. First, one identifies a combinatorial parameter, like the Littlestone or sequential fat-shattering dimension, whose finiteness provides an obvious *necessary* condition. Then, one shows that the finiteness of such a dimension is sufficient for online learnability under a suitable notion of *realizability*, where one places an assumption on the label-generating process. This step involves constructing a learning algorithm that computes the combinatorial dimension as a subroutine. These two steps were first outlined in the seminal work by Littlestone (1987). Finally, to complete the proof of sufficiency, the realizable learner is converted into an agnostic learner using the conversion introduced by Ben-David et al. (2009). By the end, the finiteness of the combinatorial dimension is established as both a necessary and sufficient condition for online learnability.

While this technique has been used to characterize online learnability for specific tuples, a general characterization for an *arbitrary* tuple $(\mathcal{X}, \mathcal{Y}, \mathcal{Z}, \mathcal{H}, \ell)$ is missing from the literature. In fact, the only known sequential complexity measure for an arbitrary learning problem is the sequential Rademacher complexity of the loss class $\ell \circ \mathcal{H} := \{(x, y) \mapsto \ell(y, h(x)) : h \in \mathcal{H}\}$. In particular, Rakhlin et al. (2015a) show that if the sequential Rademacher complexity of the loss class $\ell \circ \mathcal{H}$ is a sublinear function of $T$, then $(\mathcal{X}, \mathcal{Y}, \mathcal{Z}, \mathcal{H}, \ell)$ is online learnable. However, even for natural problems like online multiclass classification (Hanneke et al., 2023) and linear regression (Raman et al., 2024b), sublinear sequential Rademacher complexity is not *necessary* for online learnability.

**Main Contributions.** In this paper, we show that the previously outlined procedure for characterizing online learnability is *universal* - it works for any learning tuple $(\mathcal{X}, \mathcal{Y}, \mathcal{Z}, \mathcal{H}, \ell)$ as long as $\ell$ is bounded. In particular, we identify a new scale-sensitive combinatorial dimension, termed the Sequential Minimax dimension (SMdim), whose finiteness at every scale is an obvious necessary condition for online learnability. Then, by identifying the right notion of realizability and providing a new realizable-to-agnostic conversion, we establish that the finiteness of the SMdim is also sufficient for online learnability. Finally, and perhaps most surprisingly, we show that the SMdim reduces exactly to existing combinatorial dimensions in their respective setting. This includes the case where $\mathcal{Z} = \mathcal{Y}$, like the Littlestone and sequential fat-shattering dimensions, as well as the case where $\mathcal{Z} \neq \mathcal{Y}$, like the $(k+1)$-Littlestone dimension from Moran et al. (2023) and Measure shattering dimension from Raman et al. (2024a).

At the highest level of generality, the SMdim may not be insightful as it is an abstract combinatorial object that cannot be efficiently computed. However, given a specific learning problem $(\mathcal{X}, \mathcal{Y}, \mathcal{Z}, \mathcal{H}, \ell)$, one can use this object to define more concrete combinatorial objects that provide better insight into the hardness of learning and the minimax rates. In fact, in the proof of Theorem 10, our techniques illustrate how one can use tools from discrete geometry to show that existing combinatorial dimensions are just special instances of the SMdim. Thus, beyond providing a unification of existing results in online learnability, the SMdim provides a good starting point for understanding the true complexity of a learning problem $(\mathcal{X}, \mathcal{Y}, \mathcal{Z}, \mathcal{H}, \ell)$.

## 1.1. Related Works

Characterizing learnability in terms of complexity measures has a long rich history in statistical learning theory, originating from the seminal work of Vapnik and Chervonenkis (1971). In online learning, Littlestone (1987) showed that a combinatorial parameter, later named the Littlestone

dimension, provides a quantitative characterization of online binary classification in the realizable setting. Twenty-two years later, Ben-David et al. (2009) proved that the Littlestone dimension also provides a tight quantitative characterization of online binary classification in the agnostic setting. Daniely et al. (2011) generalized the Littlestone dimension to multiclass classification and showed that it fully characterizes online learnability when the label space is finite. Recently, Hanneke et al. (2023) proved that the multiclass extension of the Littlestone dimension characterizes multiclass learnability under the 0-1 loss even when the label space is unbounded. In a parallel line of work, Rakhlin et al. (2015a,b) defined the sequential fat-shattering dimension and showed that it tightly characterizes the online learnability of scalar-valued regression with respect to the absolute value loss. In addition, they defined a general complexity measure called the sequential Rademacher complexity and proved that it upper bounds the minimax expected regret of any supervised online learning game. In a similar spirit, we define a *combinatorial dimension* that upper and lower bounds the minimax expected regret of any supervised online learning game.

The proof techniques in online learning are generally constructive and result in beautiful algorithms such as Follow The (Regularized) Leader, Hedge, Multiplicative Weights, Online Gradient Descent, and so forth. In online binary classification, Littlestone (1987) proposed the Standard Optimal Algorithm and proved its optimality in the realizable setting. Daniely et al. (2011) and Rakhlin et al. (2015a) generalize this algorithm to multiclass classification and scalar-valued regression respectively. The idea of the Standard Optimal Algorithm is foundational in online learning and still appears in more recent works by Moran et al. (2023), Filmus et al. (2023), and Raman et al. (2024a). A common theme in these variants of the Standard Optimal Algorithm is their use of combinatorial dimensions to make predictions. Similarly, Rakhlin et al. (2012) use the sequential Rademacher complexity to directly construct a generic online learner in the agnostic setting. However, their online learner requires the sequential Rademacher complexity of the loss class to be sublinear in $T$, and thus does not work for arbitrary tuples $(\mathcal{X}, \mathcal{Y}, \mathcal{Z}, \mathcal{H}, \ell)$. Closing this gap, we define a new scale-sensitive dimension, named the Sequential Minimax dimension, and use it to give a generic online learner for any tuple $(\mathcal{X}, \mathcal{Y}, \mathcal{Z}, \mathcal{H}, \ell)$.

Finally, we compare our work to the recent work by Blanchard (2022) on universal online learning for bounded losses. We highlight three main differences. First, in our setup, there exists a function class $\mathcal{F}$, and the goal of the learner is to drive the expected regret with respect to $\mathcal{F}$ to 0. In contrast, there is no function class in the work by Blanchard (2022). Instead, the stream is labeled by some unknown measurable function and the goal is to drive the average cumulative loss to after placing some restrictions on the sequence of instances that can be chosen by the adversary. Second, we place no restrictions on the sequence of instances the adversary may reveal to the learner (the restriction is instead placed on how the stream is labeled). In contrast, Blanchard (2022) considers a collection of stochastic processes and restrict the adversary to play a sequence of instances sampled according to a process from this set. Finally, in our setup, the prediction space and label space may be different. In contrast, Blanchard (2022) only studies the case where the prediction and label space are the same.

## 2. Preliminaries

### 2.1. Notation

Let $\mathcal{X}$ denote the instance space, $\mathcal{Y}$ denote the label space, and $\mathcal{Z}$ denote the prediction space. For a sigma algebra $\sigma(\mathcal{Z})$ on the prediction space $\mathcal{Z}$, define $\Pi(\mathcal{Z})$ to be the set of all distributions on

$(\mathcal{Z}, \sigma(\mathcal{Z}))$. For any set $S \in \sigma(\mathcal{Z})$, let $S^c$ denote its complement. Let $\mathcal{H} \subseteq \mathcal{Z}^{\mathcal{X}}$ denote an arbitrary hypothesis class consisting of predictors $h : \mathcal{X} \to \mathcal{Z}$ that maps an instance to a prediction. Given any prediction $z \in \mathcal{Z}$ and a label $y \in \mathcal{Y}$, we consider a loss function $\ell : \mathcal{Y} \times \mathcal{Z} \to \mathbb{R}_{\geq 0}$. We put no restrictions on the loss function $\ell$, except that it is bounded, $\sup_{y,z} \ell(y, z) \leq c$ for some $c \in \mathbb{R}_{>0}$. In particular, the loss can asymmetric, and therefore we reserve the first argument for the label and the second argument for the prediction. Finally, $[N] := \{1, 2, \ldots, N\}$.

## 2.2. Supervised Online Learning

In the supervised online learning setting, an adversary plays a sequential game with the learner over $T$ rounds. In each round $t \in [T]$, the adversary selects a labeled instance $(x_t, y_t) \in \mathcal{X} \times \mathcal{Y}$ and reveals $x_t$ to the learner. The learner picks a probability measure $\mu_t \in \Pi(\mathcal{Z})$ and then makes a randomized prediction $z_t \sim \mu_t$. Finally, the adversary reveals the feedback $y_t$, and the learner suffers the loss $\ell(y_t, z_t)$. Given a hypothesis class $\mathcal{H} \subseteq \mathcal{Z}^{\mathcal{X}}$, the goal of the learner is to output randomized predictions $z_t$ such that its expected cumulative loss is close to the smallest possible cumulative loss over hypotheses in $\mathcal{H}$.

We follow the convention in online learning literature (see, e.g., Cesa-Bianchi and Lugosi (2006, Chapter 4) by defining a randomized learner as a sequence of deterministic mappings to probability distributions.

**Definition 1 (Supervised Online Learning Algorithm)** *A supervised online learning algorithm is a deterministic mapping $\mathcal{A} : (\mathcal{X} \times \mathcal{Y})^{\star} \times \mathcal{X} \to \Pi(\mathcal{Z})$ that maps past examples and the newly revealed instance $x \in \mathcal{X}$ to a probability measure $\mu \in \Pi(\mathcal{Z})$. The learner then randomly samples $z \sim \mu$ to make a prediction.*

**Remark 2** *Our definition of supervised online learning algorithm prevents an algorithm from using the realizations of its past predictions to make future predictions. While this may seem as a restriction at first, our upper bounds are achievable using online learning algorithms of exactly this type. Moreover, in Appendix A, we show that our lower bounds can be generalized to algorithms which can use past realizations of their predictions to make future plays.*

Although $\mathcal{A}$ is a deterministic mapping, the prediction $z \sim \mu$ is random. Restricting the range of $\mathcal{A}$ to be the set of Dirac measures on $\mathcal{Z}$ yields a deterministic online learner. When the context is clear, with a slight abuse of notation, we use $\mathcal{A}(x)$ to denote the random sample $z$ drawn from the distribution that $\mathcal{A}$ outputs. We say that $\mathcal{H}$ is online learnable with respect to $\ell$ if there exists an online learning algorithm $\mathcal{A}$ with "small" *expected regret*:

$$\mathrm{R}_{\mathcal{A}}(T, \mathcal{H}, \ell) := \sup_{(x_1, y_1), \ldots, (x_T, y_T)} \left( \sum_{t=1}^{T} \mathbb{E}\big[\ell(y_t, \mathcal{A}(x_t))\big] - \inf_{h \in \mathcal{H}} \sum_{t=1}^{T} \ell(y_t, h(x_t)) \right).$$

**Definition 3 (Supervised Online Learnability)** *A hypothesis class $\mathcal{H} \subseteq \mathcal{Z}^{\mathcal{X}}$ is online learnable with respect to $\ell$ if and only if $\inf_{\mathcal{A}} \mathrm{R}_{\mathcal{A}}(T, \mathcal{H}, \ell) = o(T)$.*

Implicit in our definition of expected regret and online learnability is the fact that the adversary is *oblivious* – it must pick the entire sequence of examples before the game begins. In this paper, we will always assume an oblivious adversary. That said, all our results also apply to adaptive adversaries given our definition of an online learning algorithm and the standard conversion of oblivious to adaptive regret bounds (see Exercise 4.1 in Cesa-Bianchi and Lugosi (2006)).

### 2.3. Combinatorial dimensions

In online learning theory, combinatorial dimensions play an important role in providing crisp quantitative characterizations of learnability. Formally, we define a combinatorial dimension as a function D that maps $(\mathcal{H}, \ell)$ to $\mathbb{N} \cup \{0, \infty\}$ and satisfies the following two properties: (1) $\mathcal{H}$ is online learnable with respect to $\ell$ if and only if $\mathrm{D}(\mathcal{H}, \ell) < \infty$ and (2) the minimax expected regret $\inf_{\mathcal{A}} \mathrm{R}_{\mathcal{A}}(T, \mathcal{H}, \ell)$ depends only on $\mathrm{D}(\mathcal{H}, \ell)$ and $T$. In particular, $\inf_{\mathcal{A}} \mathrm{R}_{\mathcal{A}}(T, \mathcal{H}, \ell)$ should not depend on any other property of the tuple $(\mathcal{X}, \mathcal{Y}, \mathcal{Z}, \mathcal{H}, \ell)$ such as $|\mathcal{Y}|$ or $|\mathcal{Z}|$. We also allow a combinatorial dimension to take a scale parameter as an input. That is, a scale-sensitive combinatorial dimension is a function D that maps $(\mathcal{H}, \ell)$ and a scale $\gamma > 0$ to $\mathbb{N} \cup \{0, \infty\}$ with the following two properties: (1) $\mathcal{H}$ is online learnable with respect to $\ell$ if and only if $\mathrm{D}(\mathcal{H}, \ell, \gamma) < \infty$ for every $\gamma > 0$ and (2) the minimax expected regret $\inf_{\mathcal{A}} \mathrm{R}_{\mathcal{A}}(T, \mathcal{H}, \ell)$ can be lower- and upper bounded in terms of $T$ and $\mathrm{D}(\mathcal{H}, \ell, \cdot)$. Our definition of a combinatorial dimension is similar to the definition given by Ben-David et al. (2019) with two key differences. In particular, the notion of dimension given by Ben-David et al. (2019) requires $\mathrm{D}(\mathcal{H}, \ell)$ to satisfy the finite-character property (see Section 5), but does not require it to provide a quantitative characterization of learnability.

Nevertheless, our definition of dimension also captures all existing combinatorial dimensions in online learning theory, such as the Littlestone and sequential fat-shattering dimension. These dimensions are typically defined in terms of trees, a basic combinatorial object that captures the temporal dependence inherent in online learning. Given an instance space $\mathcal{X}$ and a (potentially uncountable) set of objects $\mathcal{M}$, a $\mathcal{X}$-valued, $\mathcal{M}$-ary tree $\mathcal{T}$ of depth $T$ is a complete rooted tree such that (1) each internal node is labeled by an instance $x \in \mathcal{X}$ and (2) for every internal node and object $m \in \mathcal{M}$, there is an outgoing edge indexed by $m$. Such a tree can be identified by a sequence $(\mathcal{T}_1, \ldots, \mathcal{T}_T)$ of labeling functions $\mathcal{T}_t : \mathcal{M}^{t-1} \to \mathcal{X}$ which provide the labels for each internal node. A path of length $T$ is given by a sequence of objects $m = (m_1, \ldots, m_T) \in \mathcal{M}^T$. Then, $\mathcal{T}_t(m_1, \ldots, m_{t-1})$ gives the label of the node by following the path $(m_1, \ldots, m_{t-1})$ starting from the root node, going down the edges indexed by the $m_t$'s. We let $\mathcal{T}_1 \in \mathcal{X}$ denote the instance labeling the root node. For brevity, we define $m_{<t} = (m_1, \ldots, m_{t-1})$ and therefore write $\mathcal{T}_t(m_1, \ldots, m_{t-1}) = \mathcal{T}_t(m_{<t})$. Analogously, we let $m_{\leq t} = (m_1, \ldots, m_t)$.

Often, it is useful to label the edges of a tree with some *auxiliary* information. Given a $\mathcal{X}$-valued, $\mathcal{M}$-ary tree $\mathcal{T}$ of depth $T$ and a (potentially uncountable) set of objects $\mathcal{N}$, we can formally label the edges of $\mathcal{T}$ using objects in $\mathcal{N}$ by considering a sequence $(f_1, \ldots, f_T)$ of edge-labeling functions $f_t : \mathcal{M}^t \to \mathcal{N}$. For each depth $t \in [T]$, the function $f_t$ takes as input a path $m_{\leq t}$ of length $t$ and outputs an object in $\mathcal{N}$. Accordingly, we can think of the object $f_t(m_{\leq t})$ as labeling the edge indexed by $m_t$ after following the path $m_{<t}$ down the tree. We now use this notation to rigorously define existing combinatorial dimensions in online learning.

We start with the Littlestone dimension, which is known to characterize binary/multiclass online classification. In this setting, we take $\mathcal{Y} = \mathcal{Z}$ and $\ell(y, z) = \mathbb{1}\{y \neq z\}$.

**Definition 4 (Littlestone dimension (Littlestone, 1987; Daniely et al., 2011))** *Let $\mathcal{T}$ be a complete, $\mathcal{X}$-valued, $\{\pm 1\}$-ary tree of depth $d$. The tree $\mathcal{T}$ is shattered by $\mathcal{H} \subseteq \mathcal{Z}^{\mathcal{X}}$ if there exists a sequence $(f_1, \ldots, f_d)$ of edge-labeling functions $f_t : \{\pm 1\}^t \to \mathcal{Y}$ such that for every path $\sigma = (\sigma_1, \ldots, \sigma_d) \in \{\pm 1\}^d$, there exists a hypothesis $h_\sigma \in \mathcal{H}$ such that for all $t \in [d]$, $h_\sigma(\mathcal{T}_t(\sigma_{<t})) = f_t(\sigma_{\leq t})$ and $f_t((\sigma_{<t}, -1)) \neq f_t((\sigma_{<t}, +1))$. The Littlestone dimension of $\mathcal{H}$, denoted $\mathrm{L}(\mathcal{H})$, is the maximal depth of a tree $\mathcal{T}$ that is shattered by $\mathcal{H}$. If there exists shattered trees of arbitrarily large depth, we say $\mathrm{L}(\mathcal{H}) = \infty$.*

For online regression, where we take $\mathcal{Z} = \mathcal{Y} = [-1, 1]$ and $\ell(y, z) = |y - z|$, online learnability is characterized by the sequential-fat shattering (seq-fat) dimension.

**Definition 5 (Sequential fat-shattering dimension (Rakhlin et al., 2015a))** *Let $\mathcal{T}$ be a complete, $\mathcal{X}$-valued, $\{\pm 1\}$-ary tree of depth $d$ and fix $\gamma \in (0, 1]$. The tree $\mathcal{T}$ is $\gamma$-shattered by $\mathcal{H} \subseteq \mathcal{Z}^{\mathcal{X}}$ if there exists a sequence $(f_1, \ldots, f_d)$ of edge-labeling functions $f_t : \{\pm 1\}^t \to \mathcal{Y}$ such that for every path $\sigma = (\sigma_1, \ldots, \sigma_d) \in \{\pm 1\}^d$, there exists a hypothesis $h_\sigma \in \mathcal{H}$ such that for all $t \in [d]$, $\sigma_t(h_\sigma(\mathcal{T}_t(\sigma_{<t})) - f_t(\sigma_{\leq t})) \geq \gamma$ and $f_t((\sigma_{<t}, -1)) = f_t((\sigma_{<t}, +1))$. The sequential fat-shattering dimension of $\mathcal{H}$ at scale $\gamma$, denoted $\mathrm{sfat}_\gamma(\mathcal{H})$, is the maximal depth of a tree $\mathcal{T}$ that is $\gamma$-shattered by $\mathcal{H}$. If there exists $\gamma$-shattered trees of arbitrarily large depth, we say that $\mathrm{sfat}_\gamma(\mathcal{H}) = \infty$.*

Recently, Moran et al. (2023) study list online classification, where we take $\mathcal{Z} = \{S : S \subset \mathcal{Y}, |S| \leq k\}$ and $\ell(y, z) = \mathbb{1}\{y \notin z\}$. Here, they show that the $(k + 1)$- Littlestone dimension, characterizes online learnability of a hypothesis class $\mathcal{H} \subseteq \mathcal{Z}^{\mathcal{X}}$.

**Definition 6 ($(k + 1)$-Littlestone dimension (Moran et al., 2023))** *Let $\mathcal{T}$ be a complete, $\mathcal{X}$-valued, $[k+1]$-ary tree of depth $d$. The tree $\mathcal{T}$ is shattered by $\mathcal{H} \subseteq \mathcal{Z}^{\mathcal{X}}$ if there exists a sequence $(f_1, \ldots, f_d)$ of edge-labeling functions $f_t : [k + 1]^t \to \mathcal{Y}$ such that for every path $p = (p_1, \ldots, p_d) \in [k + 1]^d$, there exists a hypothesis $h_p \in \mathcal{H}$ such that for all $t \in [d]$, $f_t(p_{\leq t}) \in h_\sigma(\mathcal{T}_t(\sigma_{<t}))$ and for all distinct $i, j \in [k + 1]$, $f_t((p_{<t}, i)) \neq f_t((p_{<t}, j))$. The $(k + 1)$-Littlestone dimension of $\mathcal{H}$ denoted $\mathrm{L}_{k+1}(\mathcal{H})$, is the maximal depth of a tree $\mathcal{T}$ that is shattered by $\mathcal{H}$. If there exists shattered trees of arbitrarily large depth, we say that $\mathrm{L}_{k+1}(\mathcal{H}) = \infty$.*

Finally, in the "flip" of list online classification, where $\mathcal{Y} \subset \sigma(\mathcal{Z})$ is some collection of measurable subsets of $\mathcal{Z}$ and $\ell(y, z) = \mathbb{1}\{z \notin y\}$, Raman et al. (2024a) show that the Measure shattering dimension characterizes online learnability of a hypothesis class $\mathcal{H} \subseteq \mathcal{Z}^{\mathcal{X}}$.

**Definition 7 (Measure shattering dimension (Raman et al., 2024a))** *Let $\mathcal{T}$ be a complete $\mathcal{X}$-valued, $\Pi(\mathcal{Z})$-ary tree of depth $d$, and fix $\gamma \in (0, 1]$. The tree $\mathcal{T}$ is $\gamma$-shattered by $\mathcal{H} \subseteq \mathcal{Z}^{\mathcal{X}}$ if there exists a sequence $(f_1, \ldots, f_d)$ of edge-labeling set-valued functions $f_t : \Pi(\mathcal{Z})^t \to \mathcal{Y}$ such that for every path $\mu = (\mu_1, \ldots, \mu_d) \in \Pi(\mathcal{Z})^d$, there exists a hypothesis $h_\mu \in \mathcal{H}$ such that for all $t \in [d]$, $h_\mu(\mathcal{T}_t(\mu_{<t})) \in f_t(\mu_{\leq t})$ and $\mu_t(f_t(\mu_{\leq t})) \leq 1 - \gamma$. The Measure Shattering dimension (MSdim) of $\mathcal{H}$ at scale $\gamma$, denoted $\mathrm{MS}_\gamma(\mathcal{H}, \mathcal{Y})$, is the maximal depth of a tree $\mathcal{T}$ that is $\gamma$-shattered by $\mathcal{H}$. If there exists $\gamma$-shattered trees of arbitrarily large depth, we say $\mathrm{MS}_\gamma(\mathcal{H}, \mathcal{Y}) = \infty$.*

## 3. A Unifying Combinatorial Dimension

Following the procedure outlined in the introduction, we begin our characterization of online learnability by defining a dimension that provides an "obvious" necessary condition. In the context of online learning, this means giving the adversary a strategy against every possible move of the learner. Since the learner plays measures in $\Pi(\mathcal{Z})$, it suffices to consider a tree where each internal node has an outgoing edge labeled by an element of $\mathcal{Y}$ for every measure in $\Pi(\mathcal{Z})$. For any prediction $\mu \in \Pi(\mathcal{Z})$ by the learner, the label on the edge associated to $\mu$ gives the element $y \in \mathcal{Y}$ that the adversary should play to force the learner to suffer a large expected loss.

**Definition 8 (Sequential Minimax dimension)** *Let $\mathcal{T}$ be a complete $\mathcal{X}$-valued, $\Pi(\mathcal{Z})$-ary tree of depth $d$, and fix $\gamma > 0$. The tree $\mathcal{T}$ is $\gamma$-shattered by $\mathcal{H} \subseteq \mathcal{Z}^{\mathcal{X}}$ with respect to $\ell : \mathcal{Y} \times \mathcal{Z} \to \mathbb{R}_{\geq 0}$ if there exists a sequence $(f_1, \ldots, f_d)$ of edge-labeling functions $f_t : \Pi(\mathcal{Z})^t \to \mathcal{Y}$ such that for every path $\mu = (\mu_1, \ldots, \mu_d) \in \Pi(\mathcal{Z})^d$, there exists a hypothesis $h_\mu \in \mathcal{H}$ such that for all $t \in [d]$, $\mathbb{E}_{z \sim \mu_t} [\ell(f_t(\mu_{\leq t}), z)] \geq \ell(f_t(\mu_{\leq t}), h_\mu(\mathcal{T}_t(\mu_{<t}))) + \gamma$. The sequential minimax dimension (SMdim) of $\mathcal{H}$ at scale $\gamma$, denoted $\mathrm{SM}_\gamma(\mathcal{H}, \ell)$, is the maximal depth of a tree $\mathcal{T}$ that is $\gamma$-shattered by $\mathcal{H}$. If there exists $\gamma$-shattered trees of arbitrarily large depth, we say $\mathrm{SM}_\gamma(\mathcal{H}, \ell) = \infty$. Analogously, we can define $\mathrm{SM}_0(\mathcal{H}, \ell)$ by requiring strict inequality, $\mathbb{E}_{z \sim \mu} [\ell(f_t(\mu_{\leq t}), z)] > \ell(f_t(\mu_{\leq t}), h_\mu(\mathcal{T}_t(\mu_{<t})))$.*

**Remark 9** *The astute reader might notice the strong similarity between the* MSdim *and the* SMdim. *This similarity is not coincidental – the* SMdim *is a generalization of the* MSdim *designed to capture general loss functions and go beyond realizability.*

Observe that the SMdim is a function of both the hypothesis class $\mathcal{H}$ and the loss function $\ell$. However, when it is clear from context, we drop the dependence of $\ell$ and only write $\mathrm{SM}_\gamma(\mathcal{H})$. As with most scale-sensitive dimensions, SMdim has a monotonicity property, namely, $\mathrm{SM}_{\gamma_1}(\mathcal{H}) \leq \mathrm{SM}_{\gamma_2}(\mathcal{H})$ for any $\gamma_2 \leq \gamma_1$.

In Section 4, we show that the finiteness of the SMdim is both necessary and sufficient for online learnability. Theorem 10 then shows that the SMdim unifies several existing results in online supervised learning.

**Theorem 10 (Unifying Learnability)** *The following statements are true.*

(i) *If $\mathcal{Y} = \mathcal{Z}$ and $\ell(y, z) = \mathbb{1}\{y \neq z\}$, then $\mathrm{SM}_\gamma(\mathcal{H}) = \mathrm{L}(\mathcal{H})$ for all $\gamma \in [0, \frac{1}{2}]$.*

(ii) *If $\mathcal{Y} = \mathcal{Z} = [-1, 1]$ and $\ell(y, z) = |y - z|$, then $\mathrm{sfat}_\gamma(\mathcal{H}) \leq \mathrm{SM}_\gamma(\mathcal{H}) \leq \mathrm{sfat}_{\gamma'}(\mathcal{H})$ for every $0 < \gamma' < \gamma < 1$.*

(ii) *If $\mathcal{Z} = \{S : S \subset \mathcal{Y}, |S| \leq k\}$ and $\ell(y, z) = \mathbb{1}\{y \notin z\}$, then $\mathrm{SM}_\gamma(\mathcal{H}) = \mathrm{L}_{k+1}(\mathcal{H})$ for every $\gamma \in [0, \frac{1}{k+1}]$.*

(iv) *If $\mathcal{Y} \subseteq \sigma(\mathcal{Z})$ and $\ell(y, z) = \mathbb{1}\{z \notin y\}$, then $\mathrm{SM}_\gamma(\mathcal{H}) = \mathrm{MS}_\gamma(\mathcal{H}, \mathcal{Y})$ for all $y \in [0, 1]$.*

As an immediate consequence, Theorem 10 shows that the SMdim provides a tight quantitative characterization of online learnability for these problems. Our proof of Theorem 10, found in Appendix B, uses combinatorial arguments. In all four cases, our proof uses the following strategy. To show that the SMdim upper bounds the existing dimension, we take the shattered tree guaranteed by the existing dimension and for every node, use the labels on its outgoing edges to add new, labeled edges indexed by measures in $\Pi(\mathcal{Z})$. We then remove all the old edges. To show that the SMdim lower bounds the existing dimension, we take a shattered SMdim tree, and for every node, use the labels on its outgoing edges to add new, labeled edges that match the requirements of the existing dimension. Finally, we remove all the old edges indexed by measures in $\Pi(\mathcal{Z})$. In either direction, the addition of new, labeled edges requires tools from discrete geometry. For example, the proof of (ii) uses the celebrated Helly's theorem (Radon, 1921). Thus, despite being an abstract object, the techniques used in the proof of Theorem 10 show how one can use tools from discrete geometry to derive more concrete dimensions for particular choices of $(\mathcal{X}, \mathcal{Y}, \mathcal{Z}, \mathcal{H}, \ell)$.

## 4. Bounding Minimax Expected Regret

Our main result in this section shows that the finiteness of SMdim at every scale $\gamma > 0$ is both necessary and sufficient for online learnability.

**Theorem 11 (Minimax Expected Regret)**  *For any $(\mathcal{X}, \mathcal{Y}, \mathcal{Z}, \mathcal{H}, \ell)$ with $\sup_{\gamma>0} \mathrm{SM}_\gamma(\mathcal{H}) > 0$,*

$$\sup_{\gamma>0} \gamma \, \mathrm{SM}_\gamma(\mathcal{H}) \leq \inf_{\mathcal{A}} \mathrm{R}_{\mathcal{A}}(T, \mathcal{H}, \ell) \leq \inf_{\gamma>0} \left\{ \gamma T + c \, \mathrm{SM}_\gamma(\mathcal{H}) + 4c\sqrt{\mathrm{SM}_\gamma(\mathcal{H}) \, T \ln(T)} \right\}.$$

*Moreover, the upper bound and lower bound can be tight up to logarithmic factors in $T$.*

The upper bound in Theorem 9 is $o(T)$ as long as $\mathrm{SM}_\gamma(\mathcal{H}) < \infty$ for every $\gamma > 0$. Indeed, the average regret satisfies $\limsup_{T\to\infty} \inf_{\gamma>0} \left\{ \gamma + c \, \mathrm{SM}_\gamma(\mathcal{H})/T + 4c\sqrt{\mathrm{SM}_\gamma(\mathcal{H}) \, \ln(T)/T} \right\} \leq \inf_{\gamma>0} \limsup_{T\to\infty} \left\{ \gamma + c \, \mathrm{SM}_\gamma(\mathcal{H})/T + 4c\sqrt{\mathrm{SM}_\gamma(\mathcal{H}) \, \ln(T)/T} \right\} = \inf_{\gamma>0} \{\gamma\} = 0$, where the first equality follows because $\mathrm{SM}_\gamma(\mathcal{H}) < \infty, \forall \gamma > 0$.

The condition $\sup_{\gamma>0} \mathrm{SM}_\gamma(\mathcal{H}) > 0$ is necessary to ensure a non-negative lower bound. Raman et al. (2024a) provide an example of a tuple $(\mathcal{X}, \mathcal{Y}, \mathcal{Z}, \mathcal{H}, \ell)$ with $\sup_{\gamma>0} \mathrm{MS}_\gamma(\mathcal{H}) = 0$ where the corresponding minimax expected regret is negative. Moreover, Raman et al. (2024a, Example 5.1) provide a tuple $(\mathcal{X}, \mathcal{Y}, \mathcal{Z}, \mathcal{H}, \ell)$, where there exists an algorithm $\mathcal{A}$ such that $\inf_{\mathcal{A}} \mathrm{R}_{\mathcal{A}}(T, \mathcal{H}, \ell) \leq \sup_{\gamma>0} \gamma \, \mathrm{MS}_\gamma(\mathcal{H})$. Since $\mathrm{SM}_\gamma(\mathcal{H}) = \mathrm{MS}_\gamma(\mathcal{H})$ by Theorem 10, the lower bound in Theorem 11 cannot be improved in full generality. For the tightness of the upper bound, consider scalar-valued regression where $\mathcal{Y} = \mathcal{Z} = [-1, 1]$, $\ell(y, z) = |y - z|$. Since, by Theorem 10, we have that $\mathrm{SM}_\gamma(\mathcal{H}) \leq \mathrm{sfat}_{\gamma'}(\mathcal{H})$ for all $\gamma' < \gamma$, Theorem 11 implies that $\inf_{\mathcal{A}} \mathrm{R}_{\mathcal{A}}(T, \mathcal{H}, \ell) \leq \inf_{\gamma>0} \{2\gamma T + 2\mathrm{sfat}_\gamma(\mathcal{H}) + 4\sqrt{\mathrm{sfat}_\gamma(\mathcal{H}) \, T \ln(T)}\}$. However, for scalar-valued regression, Rakhlin et al. (2015a) show that $\inf_{\mathcal{A}} \mathrm{R}_{\mathcal{A}}(T, \mathcal{H}, \ell) \geq \sup_{\gamma>0} \frac{\gamma}{8}\sqrt{\mathrm{sfat}_\gamma(\mathcal{H})T}$. Thus, the upper bound in Theorem 11 is tight up to $O(\sqrt{\ln(T)})$.

The proof of Theorem 11 will follow the procedure outlined in the introduction. Namely, the lower bound will follow just from the definition of the SMdim. As for the upper bound, in Section 4.2, we will first define a notion of realizability we term $\varepsilon_t$-realizability. Then, in Lemma 12, we will constructively show that the finiteness of the SMdim at every scale is sufficient for online learnability under $\varepsilon_t$-realizability. Finally, in Section 4.3, we will provide a conversion of our $\varepsilon_t$-realizable learner into a fully agnostic online learner with the stated upper bound in Theorem 11.

### 4.1. Proof sketch of lower bound

Our proof of the lower bound in Theorem 11 is constructive. Given an algorithm and a scale $\gamma > 0$, we construct a stream by traversing the sequential minimax tree of depth $\mathrm{SM}_\gamma(\mathcal{H})$, adapting to the deterministic sequence of measures the algorithm uses to make its randomized prediction. Then, our claimed lower bound follows immediately from the definition of a shattered sequential minimax tree. Since the proof of the lower bound is relatively straightforward, we defer it to Appendix D.

### 4.2. The $\varepsilon_t$-realizable setting

In the $\varepsilon_t$-realizable setting, an adversary plays a sequential game with the learner over $T$ rounds. In each round $t \in [T]$, the adversary selects a *thresholded* labeled instance $(x_t, (y_t, \varepsilon_t)) \in \mathcal{X} \times (\mathcal{Y} \times [0, c])$ and reveals $x_t$ to the learner. The learner selects a measure $\mu_t \in \Pi(\mathcal{Z})$ and makes a randomized prediction $z_t \sim \mu_t$. Finally, the adversary reveals both the true label $y_t$ and the threshold $\varepsilon_t$ and

the learner suffers the loss $\ell(y_t, z_t)$. A sequence of thresholded labeled examples $\{(x_t, (y_t, \varepsilon_t))\}_{t=1}^T$ is called $\varepsilon_t$-realizable if there exists a hypothesis $h^\star \in \mathcal{H}$ such that $\ell(y_t, h^\star(x_t)) \leq \varepsilon_t$ for all $t \in [T]$. Given any $\varepsilon_t$-realizable stream, the goal of the learner is to output predictions such that $\sum_{t=1}^T \mathbb{1}\{\mathbb{E}_{z \sim \mu_t}[\ell(y_t, z)] \geq \gamma + \varepsilon_t\}$ is sublinear in $T$. We can think of the thresholds $\varepsilon_t$ as the adversary additionally revealing the loss that the best fixed hypothesis in hindsight suffers on the labeled instance $(x_t, y_t)$. This intuition is critical to our construction of an agnostic learner in Section 4.3. Note that if it is guaranteed ahead of time that $\varepsilon_t = 0$ for all $t \in [T]$, then this setting boils down to the standard realizable setting. Lemma 12 shows that the finiteness of $\mathrm{SM}_\gamma(\mathcal{H})$ at every scale $\gamma > 0$ is sufficient for $\mathcal{H}$ to be online learnable in $\varepsilon_t$-realizable setting.

The $\varepsilon$-additive noise setting is a widely used model in regression. The $\varepsilon$ in these works typically represents stochastic noise, which may be unbounded. In contrast, $\varepsilon_t$ is always bounded in our model. To the best of our knowledge, the $\varepsilon_t$-realizable model has not been previously studied in the learning theory literature. However, this model may provide a more realistic framework for certain practical learning scenarios and thus be of independent theoretical interest.

**Lemma 12 ($\varepsilon_t$-Realizable Learner)** *For any tuple $(\mathcal{X}, \mathcal{Y}, \mathcal{Z}, \mathcal{H}, \ell)$, Algorithm 1 running on any $\varepsilon_t$-realizable stream $\{(x_t, (y_t, \varepsilon_t))\}_{t=1}^T$ outputs $\{\mu_t\}_{t=1}^T$ such that*

$$\sum_{t=1}^T \mathbb{1}\left\{\mathbb{E}_{z \sim \mu_t}[\ell(y_t, z)] \geq \gamma + \varepsilon_t\right\} \leq \mathrm{SM}_\gamma(\mathcal{H}). \tag{1}$$

---

**Algorithm 1** Minimax Randomized Standard Optimal Algorithm (MRSOA)

---

**Input:** $\mathcal{H}$, Target accuracy $\gamma > 0$
Initialize $V_0 = \mathcal{H}$
**for** $t = 1, \ldots, T$ **do**
    Receive unlabeled example $x_t \in \mathcal{X}$.
    For all $(y, \varepsilon) \in \mathcal{Y} \times [0, c]$, define $V_{t-1}(y, \varepsilon) := \{h \in V_{t-1} \mid \ell(y, h(x_t)) \leq \varepsilon\}$.
    Define $\mathcal{C}_t := \{(y, \varepsilon) \in \mathcal{Y} \times [0, c] : |V_{t-1}(y, \varepsilon)| > 0\}$.
    If $\mathrm{SM}_\gamma(V_{t-1}) = 0$, pick $\mu_t \in \Pi(\mathcal{Z})$ such that $\mathbb{E}_{z \sim \mu_t}[\ell(y, z)] < \varepsilon + \gamma$ for all $(y, \varepsilon) \in \mathcal{C}_t$. Else,

$$\mu_t = \arg\min_{\mu \in \Pi(\mathcal{Z})} \max_{\substack{(y, \varepsilon) \in \mathcal{Y} \times [0, c] \\ \mathbb{E}_{z \sim \mu}[\ell(y, z)] \geq \varepsilon + \gamma}} \mathrm{SM}_\gamma(V_{t-1}(y, \varepsilon)).$$

    Predict $z_t \sim \mu_t$.
    Receive feedback $(y_t, \varepsilon_t)$ and update $V_t = V_{t-1}(y_t, \varepsilon_t)$.
**end**

---

To prove Lemma 12, we show that (i) on any round where $\mathbb{E}_{z_t \sim \mu_t}[\ell(y_t, z_t)] \geq \gamma + \varepsilon_t$ and $\mathrm{SM}_\gamma(V_{t-1}) > 0$, we have $\mathrm{SM}_\gamma(V_t) \leq \mathrm{SM}_\gamma(V_{t-1}) - 1$, and (ii) if $\mathrm{SM}_\gamma(V_{t-1}) = 0$ there exists a distribution $\mu_t \in \Pi(\mathcal{Z})$ such that $\mathbb{E}_{z_t \sim \mu_t}[\ell(y_t, z_t)] < \gamma + \varepsilon_t$. We defer the proof to Appendix C.

Algorithm 1 can be viewed as a generalization of RSOA introduced by Raman et al. (2024a). When $\varepsilon_t = 0$ for all $t$, then MRSOA reduces exactly to Algorithm 2 in Raman et al. (2024a). At its core, Algorithm 1 is a version space algorithm based on principles similar to that of the standard optimal algorithm (SOA) of Littlestone (1987). Recently, other variants of SOA have also been introduced in various settings. These include the Bandit SOA by Daniely and Helbertal (2013), List SOA by Moran et al. (2023), and randomized SOA by Filmus et al. (2023).

### 4.3. Realizable-to-Agnostic conversion

Now, we show how to convert Algorithm 1 into an agnostic learner satisfying the guarantee in Theorem 11. A primary approach to proving online agnostic upper bounds involves defining a set of experts that exactly covers the hypothesis class and then running multiplicative weights (Cesa-Bianchi and Lugosi, 2006) using these experts. This technique originated in work (Ben-David et al., 2009) on binary classification and was later generalized by Daniely et al. (2011) to multiclass classification. Daniely et al. (2011)'s generalization involves simulating all possible labels in $\mathcal{Y}$ to update the experts, thus making their upper bound vacuous when $|\mathcal{Y}|$ is unbounded. Recently, Hanneke et al. (2023) removed $|\mathcal{Y}|$ from the upper bound by (1) constructing an approximate cover of the hypothesis class instead of an exact cover and (2) using the feedback in the stream to update experts rather than simulating all possible labels. Our proof of the upper bound in Theorem 11 combines the ideas of both Daniely et al. (2011) and Hanneke et al. (2023). In particular, following Hanneke et al. (2023), we construct an approximate cover of the hypothesis class but follow Daniely et al. (2011) in simulating all possible *loss values*.

**Proof** (of upper bound in Theorem 11) Let $(x_1, y_1), \ldots, (x_T, y_T)$ be the data stream and $h^\star \in \arg\min_{h \in \mathcal{H}} \sum_{t=1}^{T} \ell(y_t, h(x_t))$ be an optimal function in hind-sight. For a target accuracy $\gamma > 0$, let $d_\gamma = \mathrm{SM}_\gamma(\mathcal{H})$.

**Defining Experts.** Given time horizon $T$, let $L_T = \{L \subset [T]; |L| \leq d_\gamma\}$ denote the set of all possible subsets of $[T]$ with size at most $d_\gamma$. For $\alpha \in [0, 1]$, let $\{0, \alpha, \ldots, \lceil \frac{c}{\alpha} \rceil \alpha\}$ be an $\alpha$-cover of the loss space $[0, c]$. For every $L \in L_T$, define $\Phi_L = \{0, \alpha, \ldots, \lceil \frac{c}{\alpha} \rceil \alpha\}^L$ to be the set of all functions from $L$ to the $\alpha$-cover of $[0, c]$. Given $L \in L_T$ and $\phi_L \in \Phi_L$, define an expert $E_L^{\phi_L}$ such that

$$E_L^{\phi_L}(x_t) := \mathrm{MRSOA}_\gamma\Big(x_t \mid \{i, \phi_L(i)\}_{i \in L \cap [t-1]}\Big),$$

where $\mathrm{MRSOA}_\gamma\Big(x_t \mid \{i, \phi_L(i)\}_{i \in L \cap [t-1]}\Big)$ is the prediction of the Minimax Randomized Standard Optimal Algorithm (MRSOA) running at scale $\gamma$ that has updated on thresholded labeled examples $\{(x_i, (y_i, \phi_L(i)))\}_{i \in L \cap [t-1]}$. Let $\mathcal{E} = \bigcup_{L \in L_T} \bigcup_{\phi_L \in \Phi_L} \{E_L^{\phi_L}\}$ denote the set of all Experts. Note that $|\mathcal{E}| = \sum_{i=0}^{d_\gamma} \left(\frac{2c}{\alpha}\right)^i \binom{T}{i} \leq \left(\frac{2cT}{\alpha}\right)^{d_\gamma}$.

**Multiplicative Weights as our Agnostic Learner.** Finally, given our set of experts $\mathcal{E}$, we run the Multiplicative Weights Algorithm (MWA), denoted hereinafter as $\mathcal{A}$, over the stream

$$(x_1, y_1), \ldots, (x_T, y_T)$$

with a learning rate $\eta = \sqrt{2 \ln(|\mathcal{E}|)/T}$. Let $B$ denote the random variable denoting the randomized prediction of all experts (or their corresponding randomized algorithms) . Then, conditioned on $B$, Theorem 21.11 of Shalev-Shwartz and Ben-David (2014) tells us that

$$\sum_{t=1}^{T} \mathbb{E}\left[\ell(y_t, \mathcal{A}(x_t)) \mid B\right] \leq \inf_{E \in \mathcal{E}} \sum_{t=1}^{T} \ell(y_t, E(x_t)) + c\sqrt{2T \ln(|\mathcal{E}|)} .$$

Using $|\mathcal{E}| \leq \left(\frac{2cT}{\alpha}\right)^{d_\gamma}$, and taking expectations on both sides yields

$$\mathbb{E}\left[\sum_{t=1}^{T} \ell(y_t, \mathcal{A}(x_t))\right] \leq \mathbb{E}\left[\inf_{E \in \mathcal{E}} \sum_{t=1}^{T} \ell(y_t, E(x_t))\right] + c\sqrt{2d_\gamma T \ln\left(\frac{2cT}{\alpha}\right)}. \tag{2}$$

Next, we show that the expected loss of the optimal expert is at most the loss of $h^\star$ plus a sublinear quantity.

**Tracking the Best Expert.** Define $\varepsilon_t := \ell(y_t, h^\star(x_t))$ to be the loss of the optimal hypothesis in hindsight on each round $t$. Define $\mu_t = \mu\text{-MRSOA}_\gamma\big(x_t \mid \{i, \phi_L(i)\}_{i \in L \cap [t-1]}\big)$ to be the measure returned by MRSOA$_\gamma$ (Algorithm 1) to make its randomized prediction given that the algorithm has updated on thresholded labeled examples $\{(x_i, (y_i, \phi_L(i)))\}_{i \in L \cap [t-1]}$. We say that $\mu\text{-MRSOA}_\gamma$ makes a mistake on round $t$ if $\mathbb{E}_{z_t \sim \mu_t}[\ell(y_t, z_t)] \geq \left\lceil \frac{\varepsilon_t}{\alpha} \right\rceil \alpha + \gamma$. As $\left\lceil \frac{\varepsilon_t}{\alpha} \right\rceil \alpha \geq \varepsilon_t$, the stream

$$(x_1, (y_1, \left\lceil \frac{\varepsilon_t}{\alpha} \right\rceil \alpha)), \ldots, (x_T, (y_T, \left\lceil \frac{\varepsilon_t}{\alpha} \right\rceil \alpha))$$

is $\left\lceil \frac{\varepsilon_t}{\alpha} \right\rceil \alpha$-realizable. Thus, with this notion of the mistake, Equation 1 tells us that MRSOA$_\gamma$ makes at most $d_\gamma$ mistakes on the stream $(x_1, (y_1, \left\lceil \frac{\varepsilon_t}{\alpha} \right\rceil \alpha)), \ldots, (x_T, (y_T, \left\lceil \frac{\varepsilon_t}{\alpha} \right\rceil \alpha))$.

Since $\mu\text{-MRSOA}_\gamma$ is a deterministic mapping from the past examples to a probability measure in $\Pi(\mathcal{Z})$, we can recursively define a sequence of time points where $\mu\text{-MRSOA}_\gamma$, had it run exactly on this sequence of time points, would make mistakes at each time point. To that end, let

$$t_1 = \min\left\{ t \in [T] : \mathbb{E}_{z_t \sim \mu_t}[\ell(y_t, z_t)] \geq \left\lceil \frac{\varepsilon_t}{\alpha} \right\rceil \alpha + \gamma \text{ where } \mu_t = \mu\text{-MRSOA}_\gamma\big(x_t \mid \{\}\big) \right\}$$

be the earliest time point, where a fresh, unupdated copy of $\mu\text{-MRSOA}_\gamma$ makes a mistake if it exists. Given $t_1$, we recursively define $t_i$ for $i > 1$ as

$$t_i = \min\Big\{ t > t_{i-1} : \mathbb{E}_{z_t \sim \mu_t}[\ell(y_t, z_t)] \geq \left\lceil \frac{\varepsilon_t}{\alpha} \right\rceil \alpha + \gamma,$$
$$\text{where } \mu_t = \mu\text{-MRSOA}_\gamma\left( x_t \,\Big|\, \left\{ t_j, \left\lceil \frac{\varepsilon_{t_j}}{\alpha} \right\rceil \alpha \right\}_{j=1}^{i-1} \right) \Big\}$$

if it exists. That is, $t_i$ is the earliest timepoint in $[T]$ after $t_{i-1}$ where $\mu\text{-MRSOA}_\gamma$ having updated only on the sequence $\{(x_{t_j}, (y_{t_j}, \left\lceil \frac{\varepsilon_{t_j}}{\alpha} \right\rceil \alpha))\}_{j=1}^{i-1}$ makes a mistake. We stop this process when we reach an iteration where no such time point in $[T]$ can be found where $\mu\text{-MRSOA}_\gamma$ makes a mistake.

Using the definitions above, let $t_1, t_2, \ldots,$ denote the sequence of timepoints in $[T]$ selected via this recursive procedure. Define $L^\star = \{t_1, t_2 \ldots, \}$ and $\phi_{L^\star}$ be the function such that $\phi_{L^\star}(t) = \left\lceil \frac{\varepsilon_t}{\alpha} \right\rceil \alpha$ for each $t \in L^\star$. Let $E_{L^\star}^{\phi_{L^\star}}$ be the expert parametrized by the pair $(L^\star, \phi_{L^\star})$. The expert $E_{L^\star}^{\phi_{L^\star}}$ exists because Equation (1) implies that $|L^\star| \leq d_\gamma$.

**Bounding the Loss of the Best Expert.** By definition of the expert, we have

$$E_{L^\star}^{\phi_{L^\star}}(x_t) = \text{MRSOA}_\gamma\Big( x_t \mid \{i, \phi_{L^\star}(i)\}_{i \in L^\star \cap [t-1]} \Big)$$

for all $t \in [T]$. Let us define $\mu_t^\star = \mu\text{-MRSOA}_\gamma \Big( x_t \mid \{i, \phi_{L^\star}(i)\}_{i \in L^\star \cap [t-1]} \Big)$. Using the guarantee of MRSOA (Algorithm 1), we obtain

$$
\begin{aligned}
\mathbb{E}\left[ \sum_{t=1}^{T} \ell\big(y_t, E_{L^\star}^{\phi_{L^\star}}(x_t)\big) \right] &= \sum_{t=1}^{T} \mathbb{E}_{z_t \sim \mu_t^\star}\big[ \ell\big(y_t, z_t\big) \big] \\
&\leq \sum_{t=1}^{T} c\, \mathbb{1}\left\{ \mathbb{E}_{z_t \sim \mu_t^\star}\big[ \ell\big(y_t, z_t\big) \big] \geq \left\lceil \frac{\varepsilon_t}{\alpha} \right\rceil \alpha + \gamma \right\} + \sum_{t=1}^{T} \left( \left\lceil \frac{\varepsilon_t}{\alpha} \right\rceil \alpha + \gamma \right) \\
&\leq c\, d_\gamma + \sum_{t=1}^{T} \varepsilon_t + \alpha T + \gamma T,
\end{aligned}
$$

where the final inequality uses the fact that the indicator is 1 only on $L^\star$ whose size is $\leq d_\gamma$ and $\left\lceil \frac{\varepsilon_t}{\alpha} \right\rceil \alpha \leq \varepsilon_t + \alpha$.

**Completing the Proof.** Finally, substituting this loss bound of the expert $E_{L^\star}^{\phi_{L^\star}}$ in Equation (2), we obtain

$$
\begin{aligned}
\mathbb{E}\left[ \sum_{t=1}^{T} \ell(y_t, \mathcal{A}(x_t)) \right] &\leq \sum_{t=1}^{T} \varepsilon_t + c\, d_\gamma + \alpha T + \gamma T + c\, \sqrt{ 2 d_\gamma T \ln\left( \frac{2cT}{\alpha} \right) } \\
&= \inf_{h \in \mathcal{H}} \sum_{t=1}^{T} \ell(y_t, h(x_t)) + c\, d_\gamma + \gamma T + 2c + 2c\, \sqrt{ d_\gamma T \ln(T) },
\end{aligned}
$$

where we pick $\alpha = \frac{2c}{T}$ and use the fact that $\varepsilon_t := \ell(y_t, h^\star(x_t))$. Finally, note that $c\, d_\gamma + 2c + 2c\, \sqrt{d_\gamma T \ln(T)} \leq c\, d_\gamma + 4c \sqrt{d_\gamma T \ln(T)}$. Since $\gamma > 0$ is arbitrary, this completes our proof. ∎

## 5. SMdim and the Finite Character Property

In addition to characterizing learnability, existing combinatorial dimensions in learning theory satisfy the "Finite Character Property" (FCP) (Ben-David et al., 2019; Attias et al., 2023).

**Definition 13 (Finite Character Property (Ben-David et al., 2019))** *A combinatorial dimension* $\mathrm{D}(\mathcal{H}, \ell, \gamma)$ *is said to satisfy the finite character property if for every* $d \in \mathbb{N}$ *and* $\gamma > 0$, *the statement* $\mathrm{D}(\mathcal{H}, \ell, \gamma) \geq d$ *can be demonstrated by a finite set of domain point* $X \subset \mathcal{X}$, *and a finite subset of hypotheses* $H \subset \mathcal{H}$.

In fact, according to Ben-David et al. (2019), a dimension is any function $\mathrm{D}$ that maps $(\mathcal{H}, \ell)$ to $\mathbb{N} \cup \{0, \infty\}$ and satisfies the following two properties: (1) $\mathcal{H}$ is learnable with respect to $\ell$ if and only if $\mathrm{D}(\mathcal{H}, \ell) < \infty$ and (2) $\mathrm{D}$ satisfies the FCP. This definition of dimension differs from ours since (1) it requires $\mathrm{D}$ to satisfy FCP and (2) it does not require $\mathrm{D}$ to provide a quantitative characterization.

Despite characterizing online learnability, the SMdim may not satisfy the FCP since it is defined using trees with *infinite* width. Naturally, this motivates the following question: *Under what conditions on* $(\mathcal{X}, \mathcal{Y}, \mathcal{Z}, \mathcal{H}, \ell)$ *does the SMdim satisfy the FCP?*

One way that the SMdim can satisfy the FCP is if it can be equivalently represented using trees with *finite* width. For example, in Section 3 we showed that the SMdim reduces to the Ldim,

seq-fat dimension, $(k + 1)$-Ldim, all of which are defined using finite-width trees, and thus satisfy the FCP. Additionally, we showed that SMdim reduces to MSdim from Raman et al. (2024a), who established that MSdim can be written using finite width trees when the underlying set system has a finite Helly number. A unifying property in all these settings is the fact that $(\mathcal{Y}, \mathcal{Z}, \ell)$ is a *Helly space*, a generalization of "finite dimension" to abstract spaces. More formally, given any $(\mathcal{Y}, \mathcal{Z}, \ell)$, let $\mathrm{B}_\ell(y, r) := \{z \in \mathcal{Z} : \ell(y, z) \leq r\}$ denote the "ball" of radius $r$ centered at $y$ induced by the loss $\ell$. Let $\mathrm{B}_\ell(\mathcal{Y}, \mathcal{Z}) := \{\mathrm{B}_\ell(y, r) : y \in \mathcal{Y}, r \in [0, c]\}$ to be the set of all such balls. We say $(\mathcal{Y}, \mathcal{Z}, \ell)$ is a Helly space if the *Helly number* of $\mathrm{B}_\ell(\mathcal{Y}, \mathcal{Z})$ is finite.

**Definition 14 (Helly Number)** *Let $S$ be a family of sets. The Helly number of $S$, denoted $\mathrm{H}(S)$, is the smallest number $p \in \mathbb{N}$ such that for any collection of sets $C \subseteq S$ whose intersection is empty, there is a subset $C' \subset C$ of size at most $p$ whose intersection is empty.*

The Helly number of a set system roughly quantifies the property that every sequence of sets with empty intersection has a small sub-sequence with empty intersection. In this sense, we use the Helly number of $\mathrm{B}_\ell(\mathcal{Y}, \mathcal{Z})$ to quantify a notion of "dimension" for the space $(\mathcal{Y}, \mathcal{Z}, \ell)$.

**Definition 15 (Helly Space)** *Let $\mathcal{Z} = \mathcal{Y}$. Then, we say $(\mathcal{Y}, \mathcal{Z}, \ell)$ is a Helly space if and only if $\mathrm{H}(\mathrm{B}_\ell(\mathcal{Y}, \mathcal{Z})) < \infty$. Define the Helly number of the space $(\mathcal{Y}, \mathcal{Z}, \ell)$ as $\mathrm{H}(\mathcal{Y}, \mathcal{Z}, \ell) := \mathrm{H}(\mathrm{B}_\ell(\mathcal{Y}, \mathcal{Z}))$.*

All existing work in supervised online learning theory has focused on Helly spaces. For example, in classification with the 0-1 loss, one can verify that $\mathrm{H}(\mathcal{Y}, \mathcal{Z}, \ell) = 2$. For scalar-valued regression with absolute-value loss, Helly's theorem (Radon, 1921) gives that $\mathrm{H}(\mathcal{Y}, \mathcal{Z}, \ell) = 2$. More recently, Raman et al. (2024a) showed that for online ranking with the 0-1 ranking loss, we have that $\mathrm{H}(\mathcal{Y}, \mathcal{Z}, \ell) = 2$. Online learning settings where $\mathrm{H}(\mathcal{Y}, \mathcal{Z}, \ell) \geq 3$ have also been studied. For example, in list online classification $\mathrm{H}(\mathcal{Y}, \mathcal{Z}, \ell) = k + 1$ (Moran et al., 2023). In online learning with set-valued feedback (Raman et al., 2024a), $\mathrm{H}(\mathcal{Y}, \mathcal{Z}, \ell) = \mathrm{H}(\mathcal{Y})$, where $\mathcal{Y}$ denotes an arbitrary set system defined over $\mathcal{Z}$.

Remarkably, in all of these aforementioned settings, the combinatorial dimensions that characterize learnability are defined using trees whose width is *exactly* $\mathrm{H}(\mathcal{Y}, \mathcal{Z}, \ell)$. More importantly, our proofs establishing the equivalence between the SMdim and existing combinatorial dimensions crucially utilized the Helly property of $(\mathcal{Y}, \mathcal{Z}, \ell)$ to compress the infinite width trees in the definition of SMdim to finite-width trees. These facts naturally lead to the question of whether the finiteness of $\mathrm{H}(\mathcal{Y}, \mathcal{Z}, \ell)$ provides a sufficient condition under which the SMdim can be represented using finite-width trees, and more specifically, $\mathrm{H}(\mathcal{Y}, \mathcal{Z}, \ell)$-width trees.

As an initial step towards answering this question, consider the $p$-shattering dimension defined in Definition 16. The central combinatorial object in this dimension is an $\mathcal{X}$-valued, $[p]$-ary tree $\mathcal{T}$, where $p \in \mathbb{N}$. In such a tree, each internal node of $\mathcal{T}$ has $p$ outgoing edges, where each edge is labeled by a tuple in $\mathcal{Y} \times [0, c]$. The tuple $(y, r)$ induces a ball $\mathrm{B}_\ell(y, r) := \{z \in \mathcal{Z} : \ell(y, z) \leq r\}$ in the space $(\mathcal{Y}, \mathcal{Z}, \ell)$ and we further require that the collection-wise intersection of the balls induced by the tuples labeling the $p$ edges must be empty. Such a $[p]$-ary tree is shattered by a hypothesis class if for every root-to-leaf path there exists a hypothesis whose outputs on the sequence of instances lie in the balls induced by the tuples labeling the edges along the path.

**Definition 16 ($p$-shattering dimension)** *Let $\ell : \mathcal{Z} \times \mathcal{Y} \to [0, c]$ be a loss function, $p \in \mathbb{N}$, and $\gamma > 0$. Let $\mathcal{T}$ be a complete $\mathcal{X}$-valued, $[p]$-ary tree of depth $d$. The tree $\mathcal{T}$ is $\gamma$-shattered by $\mathcal{H} \subseteq \mathcal{Z}^\mathcal{X}$ if there exists a sequence $(f_1, \ldots, f_d)$ of edge-labeling functions $f_t : [p]^t \to \mathcal{Y} \times [0, c]$ such that*

*for every path $q = (q_1, \ldots, q_d) \in [p]^d$, we have $\bigcap_{i \in [p]} \mathrm{B}\left(f_t^1((q_{<t}, i)), f_t^2((q_{<t}, i)) + \gamma\right) = \emptyset$ and there exists a hypothesis $h_q \in \mathcal{H}$ such that for all $t \in [d]$, $h_q(\mathcal{T}_t(q_{<t})) \in \mathrm{B}\left(f_t^1(q_{\leq t}), f_t^2(q_{\leq t})\right)$. The $p$-shattering dimension of $\mathcal{H}$ at scale $\gamma$, denoted $p\text{-}\dim_\gamma(\mathcal{H}, \ell)$, is the maximal depth of a tree $\mathcal{T}$ that is $\gamma$-shattered by $\mathcal{H}$. If there exists $\gamma$-shattered trees of arbitrarily large depth, we say $p\text{-}\dim_\gamma(\mathcal{H}, \ell) = \infty$.*

Note that the tree in Definition 16 is parameterized by both $p$ and $\gamma$. The number $p$ controls the width of the tree, while the number $\gamma$ is used to constrain the tuples labeling the edges. When $p = \mathrm{H}(\mathcal{Y}, \mathcal{Z}, \ell)$, the $p$-dim also reduces to all existing combinatorial dimensions in their respective setting, and thus also provides a unification of supervised online learning theory. However, unlike the SMdim, the $\mathrm{H}(\mathcal{Y}, \mathcal{Z}, \ell)$-dim is defined in terms of finite-width trees whenever $\mathrm{H}(\mathcal{Y}, \mathcal{Z}, \ell) < \infty$.

Accordingly, it is natural to ask when can the SMdim be equivalently represented using the finite-width trees in Definition 16. Lemma 17, proved in Appendix E, provides a partial answer to this question by relating the SMdim and $p$-dim whenever $(\mathcal{Y}, \mathcal{Z}, \ell)$ is a Helly space. The key intuition behind the proof of Lemma 17 is that Helly spaces allows us to effectively "compress" the infinite-width, $\Pi(\mathcal{Z})$-ary tree from the definition of SMdim to a finite-width, $[\mathrm{H}(\mathcal{Y}, \mathcal{Z}, \ell)]$-ary tree according to the definition of $p$-dim.

**Lemma 17 (SMdim $\leq p$-dim)** *For every $(\mathcal{X}, \mathcal{Y}, \mathcal{Z}, \mathcal{H}, \ell)$ such that $p^\star := \mathrm{H}(\mathcal{Y}, \mathcal{Z}, \ell) < \infty$, we have $\mathrm{SM}_\gamma(\mathcal{H}) \leq p^\star\text{-}\dim_{\gamma'}(\mathcal{H})$ for all $\gamma' < \gamma$.*

Lemma 17 implies that when $\mathrm{H}(\mathcal{Y}, \mathcal{Z}, \ell) < \infty$, the finiteness of $p^\star\text{-}\dim_\gamma(\mathcal{H})$ at every scale $\gamma$ is sufficient for online learnability. The following open question asks whether it is also necessary. *Suppose that $p^\star := \mathrm{H}(\mathcal{Y}, \mathcal{Z}, \ell) < \infty$. Does online learnability of $\mathcal{H}$ imply that $p^\star\text{-}\dim_\gamma(\mathcal{H}) < \infty$ for all $\gamma > 0$?* One way to resolve this question would be to show that $p^\star\text{-}\dim_\gamma(\mathcal{H}) \leq \mathrm{SM}_\gamma(\mathcal{H})$ for all $\gamma > 0$. A positive resolution implies that $(\mathcal{Y}, \mathcal{Z}, \ell)$ being a Helly space is a sufficient condition for SMdim to be equivalently represented using finite-width trees and therefore satisfy the FCP.

## Acknowledgments

VR acknowledges support from the NSF Graduate Research Fellowship. AT and US were supported, in part, by NSF grant DMS-2413089.

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

## Appendix A.  A More General Lower Bound

In Appendix D, we derive a lower bound on the expected regret for online learning algorithms that satisfy Definition 1. Here, we show that the same lower bound in Theorem 11 applies to a much larger family of algorithms which can also use the realizations of past plays to make future predictions. The proof is identical except now the adversary computes and uses the "expected" measure that the learner will play on round to traverse down the SM tree. We expand on this below.

In full generality, a randomized learner is a sequence of maps $f_1, f_2, \ldots, f_T$ where $f_1 : \mathcal{X} \to \Pi(\mathcal{Z})$ and $f_t : (\mathcal{X} \times \mathcal{Y})^{t-1} \times \mathcal{Z}^{t-1} \times \mathcal{X} \to \Pi(\mathcal{Z})$. On round $t$, if the learner's past predictions are $z_1, \ldots, z_{t-1}$, then its prediction on round $t$ is $z_t \sim f_t(x_{1:t-1}, y_{1:t-1}, z_{1:t-1}, x_t)$. Now, we can define the "expected" measure on round $t$ as:

$$g_t(x_{1:t-1}, y_{1:t-1}, x_t)$$
$$:= \mathbb{E}_{z_1 \sim f_1(x_1)} \left[ \mathbb{E}_{z_2 \sim f_2(x_1,y_1,z_1,x_2)} \left[ \cdots \mathbb{E}_{z_{t-1} \sim f_{t-1}(x_{1:t-2},y_{1:t-2},z_{1:t-2},x_{t-1})} \left[ f_t(x_{1:t-1}, y_{1:t-1}, z_{1:t-1}, x_t) \right] \right] \right].$$

Note that $g_t(x_{1:t-1}, y_{1:t-1}, x_t)$ is only a function of the data stream $(x_1, y_1), \ldots, (x_T, y_T)$ and so it can be computed by the adversary before the game begins. Moreover, the expected regret can be written in terms of the "expected" measures, and so our lower bounds applies to this learning algorithm if the adversary uses $g_t$'s to traverse down the SM tree as in the proof in Appendix D.

## Appendix B.  Proof of Theorem 10

In this section, we show that SMdim reduces to existing combinatorial dimensions. We start with Lemma 18, which shows that SMdim $\equiv$ Ldim.

**Lemma 18 (SMdim $\equiv$ Ldim)**  *Let $\mathcal{Y} = \mathcal{Z}$, $\mathcal{H} \subseteq \mathcal{Z}^{\mathcal{X}}$, and $\ell(y, z) = \mathbb{1}\{y \neq z\}$. Then, $\mathrm{SM}_\gamma(\mathcal{H}) = \mathrm{L}(\mathcal{H})$ for all $\gamma \in [0, \frac{1}{2}]$.*

**Proof**  Fix $\gamma \in (0, \frac{1}{2}]$. We first show that $\mathrm{SM}_\gamma(\mathcal{H}) \leq \mathrm{L}(\mathcal{H})$. Let $\mathcal{T}$ be a $\mathcal{X}$-valued, $\Pi(\mathcal{Z})$-ary tree of depth $d = \mathrm{SM}_\gamma(\mathcal{H})$ shattered by $\mathcal{H}$. Let $v$ be the root node of $\mathcal{T}$ and $x$ denote the instance labeling the node. Recall that $v$ has an outgoing edge for each measure $\mu \in \Pi(\mathcal{Z})$. Let $\{y_\mu\}_{\mu \in \Pi(\mathcal{Z})}$ be the set of elements in $\mathcal{Y}$ that label the outgoing edges from $v$. We first claim that there at least two distinct elements in the set $\{y_\mu\}_{\mu \in \Pi(\mathcal{Z})}$. For the sake of contradiction, suppose this is not the case. That is, there is only one distinct element that labels the outgoing edges from $v$. Let $y$ denote the element that labels the outgoing edges from $v$. That is, $y_\mu = y$ for all $\mu \in \Pi(\mathcal{Z})$. Consider the Dirac measure $\delta_y$ that puts all mass on $y$. Note that $\delta_y \in \Pi(\mathcal{Z})$ and therefore there exists an outgoing edge from $v$ indexed by $\delta_y$ and labeled by $y$. However, it must be the case that $\mathbb{P}_{z \sim \delta_y} [y \neq z] = 0$. Since $\gamma > 0$, the shattering condition required by Definition 8 cannot be met, which is a contradiction. Accordingly, there is at least two distinct elements in the set $\{y_\mu\}_{\mu \in \Pi(\mathcal{Z})}$.

Let $y_{-1}, y_{+1}$ be the distinct elements of the set $\{y_\mu\}_{\mu \in \Pi(\mathcal{Z})}$, and $\mu_{-1}, \mu_{+1}$ be the indices of the edges labeled by $y_{-1}$ and $y_{+1}$ respectively. Let $\mathcal{H}_{-1} = \{h_\mu : \mu \in \Pi(\mathcal{Z})^d, \mu_1 = \mu_{-1}\}$ denote the set of shattering hypothesis that corresponds to following a path down $\mathcal{T}$ that takes the outgoing edge indexed $\mu_{-1}$ from the root node. Likewise define $\mathcal{H}_{+1}$. Keep the edges indexed by $\mu_{-1}$ and $\mu_{+1}$ and remove all other outgoing edges along with their corresponding subtree. Reindex the two edges using $\{\pm 1\}$. The root node $v$ should now have two outgoing edges, indexed by $\{\pm 1\}$ and labeled by distinct elements of $\mathcal{Y}$, matching the first constraint of a Littlestone tree. As for the second constraint, observe that for all $h_{-1} \in \mathcal{H}_{-1}$ and $h_{+1} \in \mathcal{H}_{+1}$ the shattering condition from Definition 8 implies that $\mathbb{P}_{z \sim \mu_{-1}}[y_{-1} \neq z] \geq \mathbb{1}\{y_{-1} \neq h_{-1}(x)\} + \gamma$ and $\mathbb{P}_{z \sim \mu_{+1}}[y_{+1} \neq z] \geq \mathbb{1}\{y_{+1} \neq h_{+1}(x)\} + \gamma$. However, this can only be true if both $\mathbb{1}\{y_{-1} \neq h_{-1}(x)\} = 0 \implies y_{-1} = h_{-1}(x)$ and $\mathbb{1}\{y_{+1} \neq h_{+1}(x)\} = 0 \implies y_{+1} = h_{+1}(x)$. Accordingly, the hypotheses that shatters the edges indexed by $\mu_{-1}$ and $\mu_{+1}$ in the original tree according to Definition 8 also shatters the newly re-indexed edges according to Definition 4. Recursively repeating the above procedure on the subtrees following the two reindexed edges results in a Littlestone tree shattered by $\mathcal{H}$ of depth $d$. Thus, $\mathrm{SM}_\gamma(\mathcal{H}) \leq \mathrm{L}(\mathcal{H})$ for $\gamma \in (0, \frac{1}{2}]$. The case when $\gamma = 0$ follows similarly and uses the fact that when $\gamma = 0$, we define the shattering condition in SMdim with a strict inequality (see last sentence in Definition 8).

We now prove the inequality that $\mathrm{SM}_\gamma(\mathcal{H}) \geq \mathrm{L}(\mathcal{H})$. Fix $\gamma \in [0, \frac{1}{2}]$. Let $\mathcal{T}$ be a $\mathcal{X}$-valued, $\{\pm 1\}$-ary tree of depth $d = \mathrm{L}(\mathcal{H})$ shattered by $\mathcal{H}$ according to Definition 4. Our goal will be to expand $\mathcal{T}$ into a $\Pi(\mathcal{Z})$-ary tree that is $\gamma$-shattered by $\mathcal{H}$ according to Definition 8. Let $v$ be the root node of $\mathcal{T}$, $x$ be the instance that labels the root node, and $y_{-1}, y_{+1}$ denote the distinct elements of $\mathcal{Y}$ that label the left and right outgoing edges from $v$ respectively. Let $\mathcal{H}_{-1} = \{h_\sigma : \sigma \in \{\pm 1\}^d, \sigma_1 = -1\} \subset \mathcal{H}$ denote the set of shattering hypothesis that correspond to following a path down $\mathcal{T}$ that takes the edge indexed by $-1$ in the first level. Define $\mathcal{H}_{+1}$ analogously. Then, for all $h_{-1} \in \mathcal{H}_{-1}$ and $h_{+1} \in \mathcal{H}_{+1}$, the shattering condition implies that $h_{-1}(x) = y_{-1}$ and $h_{+1}(x) = y_{+1}$.

For every measure $\mu \in \Pi(\mathcal{Z})$, we claim that there exists a $\sigma_\mu \in \{\pm 1\}$ such that $\mathbb{P}_{z \sim \mu}[y_{\sigma_\mu} \neq z] = \mu(\{y_{\sigma_\mu}\}^c) \geq \gamma$. Suppose for the sake of contradiction that this is not true. Then, there exists a measure $\mu \in \Pi(\mathcal{Z})$ such that for both $\sigma \in \{\pm 1\}$, we have $\mu(\{y_\sigma\}^c) < \gamma$. Then, $1 = \mu(\mathcal{Z}) = \mu(\{y_{-1}\}^c \cup \{y_{+1}\}^c) < 2\gamma < 1$, a contradiction. Thus, for every measure $\mu \in \Pi(\mathcal{Z})$ there exists a $\sigma_\mu \in \{\pm 1\}$ such that $\mathbb{P}_{z \sim \mu}[y_{\sigma_\mu} \neq z] \geq \gamma$. Combining this with the fact that for any $h_{-1} \in \mathcal{H}_{-1}$ and $h_{+1} \in \mathcal{H}_{+1}$, we have $y_{-1} = h_{-1}(x)$ and $y_{+1} = h_{+1}(x)$, gives that, for every measure $\mu \in \Pi(\mathcal{Z})$, there exists a $\sigma_\mu \in \{\pm 1\}$ such that for all $h_{\sigma_\mu} \in \mathcal{H}_{\sigma_\mu}$, we have $\mathbb{P}_{z \sim \mu}[y_{\sigma_\mu} \neq z] \geq \mathbb{1}\{y_{\sigma_\mu} \neq h_{\sigma_\mu}(x)\} + \gamma$. Note that if we take $y_{\sigma_\mu}$ to be the label on an edge indexed by $\mu$, then the inequality above matches the shattering condition required by Definition 8.

To that end, for every measure $\mu \in \Pi(\mathcal{Z})$, add an outgoing edge from $v$ indexed by $\mu$ and labeled by the $y_{\sigma_\mu}$, where $\sigma_\mu$ is the index as promised by the analysis above. Take the sub-tree in $\mathcal{T}$ following the original outgoing edge from $v$ indexed by $\sigma_\mu$, and append it to the newly constructed outgoing edge from $v$ indexed by $\mu$. Remove the original outgoing edges from $v$ indexed by $\{\pm 1\}$ and their corresponding subtrees. Recursively repeat the above procedure on the subtrees following the newly created edges indexed by measures. Upon repeated this process for every internal node in $\mathcal{T}$, we obtain a $\Pi(\mathcal{Z})$-ary tree that is $\gamma$-shattered by $\mathcal{H}$ of depth $d$. Thus, we have that $\mathrm{L}(\mathcal{H}) \leq \mathrm{SM}_\gamma(\mathcal{H})$ for $\gamma \in [0, \frac{1}{2}]$. $\blacksquare$

Next, we show an equivalence between SMdim and seq-fat.

**Lemma 19 (SMdim ≡ seq-fat)** *Let $\mathcal{Y} = \mathcal{Z} = [-1, 1]$, $\mathcal{H} \subseteq \mathcal{Z}^{\mathcal{X}}$, and $\ell(y, z) = |y - z|$. Then for every $\gamma \in (0, 1]$ and $\gamma' < \gamma$,*

$$\mathrm{sfat}_\gamma(\mathcal{H}) \leq \mathrm{SM}_\gamma(\mathcal{H}) \leq \mathrm{sfat}_{\gamma'}(\mathcal{H}).$$

**Proof** We first prove the upper bound. Let $\gamma \in (0, 1]$ and $\gamma' < \gamma$. Let $\mathcal{T}$ be a $\mathcal{X}$-valued, $\Pi(\mathcal{Z})$-ary tree of depth $d = \mathrm{SM}_\gamma(\mathcal{H})$ shattered by $\mathcal{H}$. Let $v$ be the root node of $\mathcal{T}$ and $x$ denote the instance labeling the node. Recall that $v$ has an outgoing edge for each measure $\mu \in \Pi(\mathcal{Z})$. In particular, this means that $v$ has outgoing edges corresponding to the Dirac measures on $\mathcal{Z}$, which we denote by $\{\delta_z\}_{z \in \mathcal{Z}}$. Fix a $z \in \mathcal{Z}$ and consider the outgoing edge from $v$ indexed by $\delta_z$. Let $y_z \in \mathcal{Y}$ be the element that labels the outgoing edge indexed by $\delta_z$. Let $\mathcal{H}_z = \{h_\mu : \mu \in \Pi(\mathcal{Z})^d, \mu_1 = \delta_z\} \subset \mathcal{H}$ denote the set of shattering hypothesis that corresponds to following a path down $\mathcal{T}$ that takes the edge $\delta_z$ in the root node. Then, for all $h \in \mathcal{H}_z$ the shattering condition from Definition 8 implies that

$$|z - y_z| \geq |h(x) - y_z| + \gamma > |h(x) - y_z| + \gamma'.$$

Taking the supremum on both sides, gives that:

$$|z - y_z| > \sup_{h \in \mathcal{H}_z} |h(x) - y_z| + \gamma' = r_z + \gamma'. \tag{3}$$

where we let $r_z = \sup_{h \in \mathcal{H}_z} |h(x) - y_z|$. Let $I_z := [y_z - (r_z + \gamma'), y_z + (r_z + \gamma')] \subset [-3, 3]$ denote an interval corresponding to $z$. Inequality (3) above implies that $z \notin I_z$ (note that $I_z$ changes depending on $z$). Since $z \in \mathcal{Z}$ was arbitrary, it must be the case that $z \notin I_z$ for all $z \in \mathcal{Z}$. This means that $\bigcap_{z \in \mathcal{Z}} I_z = \emptyset$. Since $[-3, 3]$ is compact and $\{I_z\}_{z \in \mathcal{Z}}$ is a family of closed intervals whose intersection is empty, the celebrated Helly's theorem states that there exists two intervals in $\{I_z\}_{z \in \mathcal{Z}}$ that are disjoint (Eckhoff, 1993; Radon, 1921). Accordingly, let $z_1, z_2$ be such that $I_{z_1} \cap I_{z_2} = \emptyset$. As before, let $y_{z_1}$ and $y_{z_2}$ be the labels on the outgoing edges from $v$ indexed by the Dirac measures $\delta_{z_1}$ and $\delta_{z_2}$ respectively. Without loss of generality, let $y_{z_1} < y_{z_2}$ (we have strict inequality because we are guaranteed that $I_{z_1}$ and $I_{z_2}$ are disjoint). By inequality 3, for all $h_{z_1} \in \mathcal{H}_{z_1}$ and $h_{z_2} \in \mathcal{H}_{z_2}$ we have that

$$h_{z_1}(x) \in [y_{z_1} - r_{z_1}, y_{z_1} + r_{z_1}] \quad \text{and} \quad h_{z_2}(x) \in [y_{z_2} - r_{z_2}, y_{z_2} + r_{z_2}].$$

Let $s = \frac{y_{z_1} + r_{z_1} + y_{z_2} - r_{z_2}}{2} \in [-1, 1]$ be a witness. Then, for all $h_{z_1} \in \mathcal{H}_{z_1}$ and $h_{z_2} \in \mathcal{H}_{z_2}$, we have that $s - h_{z_1}(x) \geq \gamma'$ and $h_{z_2}(x) - s \geq \gamma'$. Relabel the two edges indexed by $\delta_{z_1}$ and $\delta_{z_2}$ with the same witness $s$. Reindex the two edges indexed by $\delta_{z_1}$ and $\delta_{z_2}$ with $-1$ and $+1$ respectively. Remove all other edges indexed by measures and their corresponding subtrees. There should now only be two outgoing edges from $v$, each labeled by the same witness. Next, recall that for all $h_{z_1} \in \mathcal{H}_{z_1}$ and $h_{z_2} \in \mathcal{H}_{z_2}$ we have that $s - h_{z_1} \geq \gamma'$ and $h_{z_2} - s \geq \gamma'$. Accordingly, the hypotheses that shatter the edges indexed by $\delta_{z_1}$ and $\delta_{z_2}$ in the original tree according to Definition 8 also shatter the newly re-indexed and relabeled edges according to Definition 5. Recursively repeating the above procedure on the subtrees following the two newly reindexed and relabeled edges results in a seq-fat tree $\gamma'$-shattered by $\mathcal{H}$ of depth $d$. Thus, $\mathrm{SM}_\gamma(\mathcal{H}) \leq \mathrm{sfat}_{\gamma'}(\mathcal{H})$ for $\gamma' < \gamma$.

We now move on to prove the lower bound. Let $\gamma \in (0, 1]$ and $\mathcal{T}$ be a $\mathcal{X}$-valued, $\{\pm 1\}$-ary tree of depth $d = \mathrm{sfat}_\gamma(\mathcal{H})$ shattered by $\mathcal{H}$ according to Definition 5. Our goal will be expand $\mathcal{T}$

into a $\Pi(\mathcal{Z})$-ary tree that is $\gamma$-shattered by $\mathcal{H}$ according to Definition 8. Let $v$ be the root node, $x$ the instance that labels the root node, and $s$ be the witness that labels the two outgoing edges of $v$. Let $\mathcal{H}_{-1} = \{h_\sigma : \sigma \in \{\pm 1\}^d, \sigma_1 = -1\} \subset \mathcal{H}$ denote the set of shattering hypothesis that corresponds to following a path down $\mathcal{T}$ that takes the outgoing edge indexed by $-1$ from the root node. Likewise define $\mathcal{H}_{+1}$. Then, for all $h_{-1} \in \mathcal{H}_{-1}$ and $h_{+1} \in \mathcal{H}_{+1}$, the shattering condition implies that $s - h_{-1}(x) \geq \gamma$ and $h_{+1}(x) - s \geq \gamma$ respectively.

For every measure $\mu \in \Pi(\mathcal{Z})$, we claim that there exist a $\sigma_\mu \in \{-1, 1\}$ such that $\mathbb{E}_{z \sim \mu}[|\sigma_\mu - z|] \geq |s - \sigma_\mu|$. Suppose for the sake of contradiction that this is not true. That is, there exists $\mu \in \Pi(\mathcal{Z})$ such that for all $\tau \in \{-1, 1\}$ we have that $\mathbb{E}_{z \sim \mu}[|\tau - z|] < |s - \tau|$. Then, when $\tau = -1$, we have that $\mathbb{E}_{z \sim \mu}[z] < |s + 1| - 1$ and when $\tau = 1$, we have $1 - |s - 1| < \mathbb{E}_{z \sim \mu}[z]$, using the fact that $|\tau - z| = 1 - \tau z$. Combining the two inequalities together and using the fact that $s \in [-1, 1]$ gives that $2 < |s + 1| + |s - 1| = 2$, which is a contradiction. Accordingly, for every measure $\mu \in \Pi(\mathcal{Z})$, there exists a $\sigma_\mu \in \{-1, 1\}$ such that $\mathbb{E}_{z \sim \mu}[|\sigma_\mu - z|] \geq |s - \sigma_\mu|$. Next, crucially note that for any $\tau \in \{\pm 1\}$ and any $h_\tau \in \mathcal{H}_\tau$, we have $|h_\tau(x) - \tau| = |s - \tau| - |h_\tau(x) - s| \leq |s - \tau| - \gamma$ by the seq-fat shattering condition from Definition 5. Therefore, for every measure $\mu \in \Pi(\mathcal{Z})$, there exists $\sigma_\mu \in \{\pm 1\}$ such that for all $h_{\sigma_\mu} \in \mathcal{H}_{\sigma_\mu}$, we have that $\mathbb{E}_{z \sim \mu}[|\sigma_\mu - z|] \geq |\sigma_\mu - h_{\sigma_\mu}(x)| + \gamma$. Note that if we take $\sigma_\mu$ to be the label on a edge indexed by $\mu$, then $\mathbb{E}_{z \sim \mu}[|\sigma_\mu - z|] \geq |\sigma_\mu - h_{\sigma_\mu}(x)| + \gamma$ exactly matches the shattering condition required by Definition 8.

To that end, for every measure $\mu \in \Pi(\mathcal{Z})$, add an outgoing edge from $v$ indexed by $\mu$ and labeled by the $\sigma_\mu \in \{\pm 1\}$ promised in the analysis above. Take the sub-tree in $\mathcal{T}$ following the original outgoing edge from $v$ indexed by $\sigma_\mu$, and append it to the newly constructed outgoing edge from $v$ indexed by $\mu$. Remove the original outgoing edges from $v$ indexed by $-1$ and $+1$ and their corresponding subtrees. Recursively repeat the above procedure on the subtrees following the newly created edges indexed by measures. Upon repeating this process for every internal node in $\mathcal{T}$, we obtain a $\Pi(\mathcal{Z})$-ary tree that is $\gamma$-shattered by $\mathcal{H}$ of depth $d$. Thus, we have that $\mathrm{sfat}_\gamma(\mathcal{H}) \leq \mathrm{SM}_\gamma(\mathcal{H})$. ∎

Next, we show that SMdim reduces to $(k + 1)$-Ldim from Moran et al. (2023).

**Lemma 20 (SMdim $\equiv (k+1)$-Ldim)** *Let $\mathcal{Z} = \{S : S \subset \mathcal{Y}, |S| \leq k\}$, $\mathcal{H} \subseteq \mathcal{Z}^{\mathcal{X}}$, and $\ell(y, z) = \mathbb{1}\{y \notin z\}$. Then for every $\gamma \in [0, \frac{1}{k+1}]$, we have $\mathrm{SM}_\gamma(\mathcal{H}) = \mathrm{L}_{k+1}(\mathcal{H})$.*

**Proof** Fix $\gamma \in (0, 1]$. We first show that $\mathrm{SM}_\gamma(\mathcal{H}) \leq \mathrm{L}_{k+1}(\mathcal{H})$. Let $\mathcal{T}$ be a $\mathcal{X}$-valued, $\Pi(\mathcal{Z})$-ary tree of depth $d = \mathrm{SM}_\gamma(\mathcal{H})$ shattered by $\mathcal{H}$. Let $v$ be the root node of $\mathcal{T}$ and $x$ denote the instance labeling the node. Recall that $v$ has an outgoing edge for each measure $\mu \in \Pi(\mathcal{Z})$. Let $\{y_\mu\}_{\mu \in \Pi(\mathcal{Z})}$ be the set of elements in $\mathcal{Y}$ that label the outgoing edges from $v$. We first claim that there at least $k + 1$ distinct elements in the set $\{y_\mu\}_{\mu \in \Pi(\mathcal{Z})}$. For the sake of contradiction, suppose this was not the case. That is, there are only $k$ distinct elements that label the outgoing edges from $v$. Let $y_1, \ldots, y_k$ denote the $k$ distinct elements that label the outgoing edges from $v$, Consider the measure $\tilde{\mu}$ that puts all mass on $\{y_1, \ldots, y_k\}$. Note that $\tilde{\mu} \in \Pi(\mathcal{Z})$ and let $\tilde{y} \in \{y_1, \ldots, y_k\}$ be the label on the outgoing edge from $v$ indexed by $\tilde{\mu}$. By definition of $\tilde{\mu}$ and $\tilde{y}$, it must be the case that $\mathbb{P}_{z \sim \tilde{\mu}}[\tilde{y} \notin z] = 0$. Since $\gamma > 0$, the shattering condition required by Definition 8 cannot be met, which is a contradiction. Accordingly, there exists at least $k + 1$ distinct elements in the set $\{y_\mu\}_{\mu \in \Pi(\mathcal{Z})}$.

Let $y_1, \ldots, y_{k+1}$ be the distinct elements of the set $\{y_\mu\}_{\mu \in \Pi(\mathcal{Z})}$, and $\mu_1, \ldots, \mu_{k+1}$ be the indices of the edges labeled by $y_1, \ldots, y_{k+1}$ respectively, breaking ties arbitrarily. For $\mu_i \in \{\mu_1, \ldots, \mu_{k+1}\}$, let $\mathcal{H}_{\mu_i}$ denote the set of shattering hypothesis that corresponds to following a path down $\mathcal{T}$ that

takes the outgoing edge $\mu_i$ from the root node. Keep the edges indexed by $\mu_1, \ldots, \mu_{k+1}$, and re-move all other outgoing edges along with their corresponding subtree. Reindex the $k + 1$ edges using distinct numbers in $[k + 1]$. The root node $v$ should now have $k + 1$ outgoing edges, each indexed by a different natural number in $[k + 1]$ and labeled by a distinct element of $\mathcal{Y}$, matching the first constraint of a $(k + 1)$-Littlestone tree. As for the second constraint, observe that for all $h \in \mathcal{H}_{\mu_i}$ the shattering condition implies that $\mathbb{P}_{z \sim \mu_i}[y_i \notin z] \geq \mathbb{1}\{y_i \notin h(x)\} + \gamma$. However, this can only be true if $\mathbb{1}\{y_i \notin h(x)\} = 0 \implies y_i \in h(x)$. Accordingly, the hypotheses that shatter the edges indexed by $\mu_1, \ldots, \mu_{k+1}$ in the original tree according to Definition 8 also shatter the newly re-indexed edges according to Definition 6. Recursively repeating the above procedure on the subtrees following the $k + 1$ reindexed edges results in a $(k + 1)$-Littlestone tree shattered by $\mathcal{H}$ of depth $d$. Thus, $\mathrm{SM}_\gamma(\mathcal{H}) \leq \mathrm{L}_{k+1}(\mathcal{H})$ for $\gamma \in (0, 1]$. The case when $\gamma = 0$ follows similarly and uses the fact that when $\gamma = 0$, we define the shattering condition in SMdim with a strict inequality (see last sentence in Definition 8).

We now prove the inequality that $\mathrm{SM}_\gamma(\mathcal{H}) \geq \mathrm{L}_{k+1}(\mathcal{H})$. Fix $\gamma \in [0, \frac{1}{k+1}]$. Let $\mathcal{T}$ be a $\mathcal{X}$-valued, $[k + 1]$-ary tree of depth $d = \mathrm{L}_{k+1}(\mathcal{H})$ shattered by $\mathcal{H}$ according to Definition 6. Our goal will be to expand $\mathcal{T}$ into a $\Pi(\mathcal{Z})$-ary tree that is $\gamma$-shattered by $\mathcal{H}$ according to Definition 8. Let $v$ be the root node of $\mathcal{T}$, $x$ be the instance that labels the root node, and $\{y_i\}_{i=1}^{k+1}$ denote the distinct elements of $\mathcal{Y}$ that label the $k + 1$ outgoing edges from $v$. For each $i \in [k + 1]$, let $\mathcal{H}_i = \{h_p : p \in [k + 1]^d, p_1 = i\} \subset \mathcal{H}$ denote the set of shattering hypothesis that corresponds to following a path down $\mathcal{T}$ that takes the outgoing edge indexed by $i$ from $v$. Then, for all $i \in [k + 1]$ and $h_i \in \mathcal{H}_i$, the shattering condition implies that $y_i \in h_i(x) \implies \mathbb{1}\{y_i \notin h_i(x)\} = 0$.

For every measure $\mu \in \Pi(\mathcal{Z})$, we claim that there exists a $i_\mu \in [k+1]$ such that $\mathbb{P}_{z \sim \mu}[y_{i_\mu} \notin z] \geq \gamma$. Suppose for the sake of contradiction that this is not true. Then, there exists a measure $\mu \in \Pi(\mathcal{Z})$ such that for all $i \in [k + 1]$, we have $\mathbb{P}_{z \sim \mu}[y_i \notin z] < \gamma$. This implies that

$$\mathbb{P}_{z \sim \mu}[\exists i \in [k + 1] \text{ such that } y_i \notin z] < (k + 1)\gamma < 1.$$

However, since $\mu$ is supported over subsets of $\mathcal{Y}$ of size $\leq k$, we have $\mathbb{P}_{z \sim \mu}[\exists i \in [k + 1] \text{ such that } y_i \notin z] = 1$, a contradiction. Thus, for every measure $\mu \in \Pi(\mathcal{Z})$ there exists a $i_\mu \in [k + 1]$ such that $\mathbb{P}_{z \sim \mu}[y_{i_\mu} \notin z] \geq \gamma$. Combining this with the fact that for every $i \in [k + 1]$ and $h_i \in \mathcal{H}_i$ we have that $y_i \in h_i(x)$ gives that, for every measure $\mu \in \Pi(\mathcal{Z})$, there exists a $i_\mu \in [k + 1]$ such that for all $h_{i_\mu} \in \mathcal{H}_{i_\mu}$, we have $\mathbb{P}_{z \sim \mu}[y_{i_\mu} \notin z] \geq \mathbb{1}\{y_{i_\mu} \notin h_{i_\mu}(x)\} + \gamma$. Note that if we take $y_{i_\mu}$ to be the label on an edge indexed by $\mu$, then the inequality above matches the shattering condition required by Definition 8.

To that end, for every measure $\mu \in \Pi(\mathcal{Z})$, add an outgoing edge from $v$ indexed by $\mu$ and labeled by the $y_{i_\mu}$, where $i_\mu$ is the index as promised by the analysis above. Take the sub-tree in $\mathcal{T}$ following the original outgoing edge from $v$ indexed by $i_\mu$, and append it to the newly constructed outgoing edge from $v$ indexed by $\mu$. Remove the original outgoing edges from $v$ indexed by numbers in $[k + 1]$ and their corresponding subtrees. Recursively repeat the above procedure on the subtrees following the newly created edges indexed by measures. Upon repeating this process for every internal node in $\mathcal{T}$, we obtain a $\Pi(\mathcal{Z})$-ary tree of depth $d$ that is $\gamma$-shattered by $\mathcal{H}$. Thus, we have that $\mathrm{L}_{k+1}(\mathcal{H}) \leq \mathrm{SM}_\gamma(\mathcal{H})$ for $\gamma \in [0, \frac{1}{k+1}]$. $\blacksquare$

Finally, we show that the SMdim $\equiv$ MSdim.

**Lemma 21 (SMdim $\equiv$ MSdim)** *Let $\mathcal{Y} \subset \sigma(\mathcal{Z})$, $\mathcal{H} \subseteq \mathcal{Z}^{\mathcal{X}}$, and $\ell(y, z) = \mathbb{1}\{z \notin y\}$. Then for every $\gamma \in [0, 1]$, we have $\mathrm{SM}_\gamma(\mathcal{H}) = \mathrm{MS}_\gamma(\mathcal{H})$.*

**Proof** The equality follows directly from the fact that $\mathbb{E}_{z \sim \mu}\left[\ell(y, z)\right] = \mu(y^c)$ and the fact that $\mathbb{E}_{z \sim \mu_t}\left[\ell(z, f_t(\mu_{\leq t}))\right] \geq \ell(h_\mu(\mathcal{T}_t(\mu_{<t})), f_t(\mu_{\leq t})) + \gamma \iff h_\mu(\mathcal{T}_t(\mu_{<t})) \in f_t(\mu_{\leq t})$ and $\mu_t(f_t(\mu_{\leq t})) \leq 1 - \gamma$. ∎

## Appendix C. Proof of Lemma 12

We now prove that given any target accuracy $\gamma > 0$ and any $\varepsilon_t$-realizable sequence $\{(x_t, (y_t, \varepsilon_t))\}_{t=1}^T$, Algorithm 1 computes distributions $\mu_t \in \Pi(\mathcal{Z})$ such that

$$\sum_{t=1}^T \mathbb{1}\{\mathbb{E}_{z \sim \mu_t}\left[\ell(y_t, z)\right] \geq \gamma + \varepsilon_t\} \leq \mathrm{SM}_\gamma(\mathcal{H}).$$

To prove this guarantee, it suffices to show that (i) on any round where $\mathbb{E}_{z_t \sim \mu_t}\left[\ell(y_t, z_t)\right] \geq \gamma + \varepsilon_t$ and $\mathrm{SM}_\gamma(V_{t-1}) > 0$, we have $\mathrm{SM}_\gamma(V_t) \leq \mathrm{SM}_\gamma(V_{t-1}) - 1$, and (ii) if $\mathrm{SM}_\gamma(V_{t-1}) = 0$ there always exists a distribution $\mu_t \in \Pi(\mathcal{Z})$ such that $\mathbb{E}_{z_t \sim \mu_t}\left[\ell(y_t, z_t)\right] < \gamma + \varepsilon_t$.

Let $t \in [T]$ be a round where $\mathbb{E}_{z_t \sim \mu_t}\left[\ell(y_t, z_t)\right] \geq \gamma + \varepsilon_t$ and $\mathrm{SM}_\gamma(V_{t-1}) > 0$. For the sake contradiction, suppose that $\mathrm{SM}_\gamma(V_t) = \mathrm{SM}_\gamma(V_{t-1}) = d$. Then, by the min-max computation in Algorithm 1, for every measure $\mu \in \Pi(\mathcal{Z})$, there exists a pair $(y_\mu, \varepsilon_\mu) \in \mathcal{Y} \times [0, c]$ such that $\mathbb{E}_{z \sim \mu}\left[\ell(y_\mu, z)\right] \geq \varepsilon_\mu + \gamma$ and $\mathrm{SM}_\gamma(V_{t-1}(y_\mu, \varepsilon_\mu)) = d$. Now construct a tree $\mathcal{T}$ with $x_t$ labeling the root node. For each measure $\mu \in \Pi(\mathcal{Z})$, construct an outgoing edge from $x_t$ indexed by $\mu$ and labeled by $y_\mu$. Append the tree of depth $d$ associated with the version space $V_{t-1}(y_\mu, \varepsilon_\mu)$ to the edge indexed by $\mu$. Note that the depth of $\mathcal{T}$ must be $d + 1$. Furthermore, observe that for every hypothesis $h \in V_{t-1}(y_\mu, \varepsilon_\mu)$, we have that $\mathbb{E}_{z \sim \mu}\left[\ell(y_\mu, z)\right] \geq \ell(y_\mu, h(x_t)) + \gamma$, matching the shattering condition in Definition 8. Therefore, by definition of SMdim, we have that $\mathrm{SM}_\gamma(V_{t-1}) \geq d + 1$, a contradiction. Thus, it must be the case that $\mathrm{SM}_\gamma(V_t) \leq \mathrm{SM}_\gamma(V_{t-1}) - 1$.

Now, suppose $t \in [T]$ is a round such that $\mathrm{SM}_\gamma(V_{t-1}) = 0$. We show that there always exist a distribution $\mu_t \in \Pi(\mathcal{Z})$ such that for all $(y, \varepsilon) \in \mathcal{C}_t$, we have $\mathbb{E}_{z_t \sim \mu_t}\left[\ell(y, z_t)\right] < \gamma + \varepsilon$. Since we are in the $\varepsilon_t$-realizable setting, it must be the case that $(y_t, \varepsilon_t) \in \mathcal{C}_t$. To see why such a $\mu_t$ must exist, suppose for the sake of contradiction that it does not exist. Then, for all $\mu \in \Pi(\mathcal{Z})$, there exists a pair $(y_\mu, \varepsilon_\mu) \in \mathcal{C}_t$ such that $\mathbb{E}_{z \sim \mu}\left[\ell(y_\mu, z)\right] \geq \gamma + \varepsilon_\mu$. As before, consider a tree with root node labeled by $x_t$. For each measure $\mu \in \Pi(\mathcal{Z})$, construct an outgoing edge from $x_t$ indexed by $\mu$ and labeled by $y_\mu$. Since $(y_\mu, \varepsilon_\mu) \in \mathcal{C}_t$, there exists a hypothesis $h_\mu \in V_{t-1}$ such that $\ell(y_\mu, h_\mu(x_t)) \leq \varepsilon_\mu$. Therefore, we have $\mathbb{E}_{z \sim \mu}\left[\ell(y_\mu, z)\right] \geq \ell(y_\mu, h_\mu(x_t)) + \gamma$. By definition of SMdim, this implies that $\mathrm{SM}_\gamma(V_{t-1}) \geq 1$, which contradicts the fact that $\mathrm{SM}_\gamma(V_{t-1}) = 0$. Thus, there must be a distribution $\mu_t \in \Pi(\mathcal{Z})$ such that for for all $(y, \varepsilon) \in \mathcal{C}_t$, we have $\mathbb{E}_{z \sim \mu_t}\left[\ell(y, z)\right] < \gamma + \varepsilon$. Since this is precisely the distribution that Algorithm 1 plays whenever $\mathrm{SM}_\gamma(V_{t-1}) = 0$ and since $\mathrm{SM}_\gamma(V_{t'}) \leq \mathrm{SM}_\gamma(V_{t-1})$ for all $t' \geq t$, the algorithm no longer suffers expected loss more than $\gamma + \varepsilon_{t'}$ for all $t' \geq t$. This completes the proof.

## Appendix D. Proof of lower bound in Theorem 11

We now prove the lower bound in Theorem 11. Fix $\gamma > 0$ and $d_\gamma := \mathrm{SM}_\gamma(\mathcal{H})$. By definition of SMdim, there exists a $\mathcal{X}$-valued, $\Pi(\mathcal{Z})$-ary tree $\mathcal{T}$ of depth $d_\gamma$ shattered by $\mathcal{H}$. Let $(f_1, \ldots, f_d)$ be the sequence of edge-labeling functions $f_t : \Pi(\mathcal{Z})^t \to \mathcal{Y}$ associated with $\mathcal{T}$. Let $\mathcal{A}$ be any randomized learner for $\mathcal{H}$. Our goal will be to use $\mathcal{T}$ and its edge-labeling functions $(f_1, \ldots, f_d)$

to construct a difficult stream for $\mathcal{A}$ such that on every round, the expected loss of $\mathcal{A}$ is at least $\gamma$ more than the loss of the optimal hypothesis in hindsight. This stream is obtained by traversing $\mathcal{T}$ adapting to the sequence of distributions output by $\mathcal{A}$.

To that end, for every round $t \in [d_\gamma]$, let $\mu_t$ denote the distribution that $\mathcal{A}$ computes before making its prediction $z_t \sim \mu_t$. Consider the stream $\{(\mathcal{T}_t(\mu_{<t}), f_t(\mu_{\leq t}))\}_{t=1}^{d_\gamma}$, where $\mu = (\mu_1, \ldots, \mu_{d_\gamma})$ denotes the sequence of distributions output by $\mathcal{A}$. This stream is obtained by starting at the root of $\mathcal{T}$, passing $\mathcal{T}_1$ to $\mathcal{A}$, observing the distribution $\mu_1$ computed by $\mathcal{A}$, passing the label $f_t(\mu_{\leq 1})$ to $\mathcal{A}$, and then finally moving along the edge indexed by $\mu_1$. This process then repeats $d_\gamma - 1$ times until the end of the tree $\mathcal{T}$ is reached. Note that we can observe and use the distribution computed by $\mathcal{A}$ on round $t$ to generate the label because $\mathcal{A}$ *deterministically* maps a sequence of labeled instances to a distribution.

Recall that the shattering condition implies that $\exists h_\mu \in \mathcal{H}$ such that $\mathbb{E}_{z_t \sim \mu_t}[\ell(f_t(\mu_{\leq t}), z_t)] \geq \ell(f_t(\mu_{\leq t}), h_\mu(\mathcal{T}_t(\mu_{<t}))) + \gamma$ for all $t \in [d_\gamma]$. Therefore, the regret of $\mathcal{A}$ on the stream described above is at least

$$\mathrm{R}_\mathcal{A}(T, \mathcal{H}, \ell) \geq \sum_{t=1}^{d_\gamma} \mathbb{E}_{z_t \sim \mu_t}[\ell(f_t(\mu_{\leq t}), z_t)] - \sum_{t=1}^{d_\gamma} \ell(f_t(\mu_{\leq t}), h_\mu(\mathcal{T}_t(\mu_{<t}))) \geq \sum_{t=1}^{d_\gamma} \gamma = \gamma d_\gamma.$$

Since our choice of $\gamma$ and the randomized algorithm $\mathcal{A}$ is arbitrary, this holds true for any $\gamma > 0$ and randomized online learner. This completes our proof.

## Appendix E. Proof of Lemma 17

Let $p = \mathrm{H}(\mathcal{Y}, \mathcal{Z}, \ell)$. Fix $\gamma \in (0, 1]$ and $\gamma' < \gamma$. Let $\mathcal{T}$ be a $\mathcal{X}$-valued, $\Pi(\mathcal{Z})$-ary tree of depth $d = \mathrm{SM}_\gamma(\mathcal{H})$ shattered by $\mathcal{H}$. Let $v$ be the root node of $\mathcal{T}$ and $x$ denote the instance labeling $v$. Recall that $v$ has an outgoing edge for each measure $\mu \in \Pi(\mathcal{Z})$. In particular, this means that $v$ has outgoing edges corresponding to the Dirac measures on $\mathcal{Z}$, which we denote by $\{\delta_z\}_{z \in \mathcal{Z}}$. Fix a $z \in \mathcal{Z}$ and consider the outgoing edge from $v$ indexed by $\delta_z$. Let $y_z \in \mathcal{Y}$ be the element that labels the outgoing edge indexed by $\delta_z$. Let $\mathcal{H}_{\delta_z} = \{h_\mu : \mu \in \Pi(\mathcal{Z})^d, \mu_1 = \delta_z\} \subset \mathcal{H}$ denote the set of shattering hypothesis that corresponds to following a path down $\mathcal{T}$ that takes the edge $\delta_z$ in the first level. Then, for all $h \in \mathcal{H}_\delta$, the shattering condition implies that

$$\ell(y_z, z) \geq \ell(y_z, h(x)) + \gamma > \ell(y_z, h(x)) + \gamma'.$$

Taking the supremum on both sides further gives that

$$\ell(y_z, z) > \sup_{h \in \mathcal{H}_{\delta_z}} \ell(y_z, h(x)) + \gamma'.$$

Let $B_z := \mathrm{B}_\ell(y_z, r_z + \gamma') \in \mathrm{B}_\ell(\mathcal{Y})$ be the ball centered around $y_z$ of radius $r_z + \gamma'$ where $r_z := \sup_{h \in \mathcal{H}_{\delta_z}} \ell(y_z, h(x))$. The inequality above implies that $z \notin B_z$ (note that $B_z$ changes depending on $z$). Since $z \in \mathcal{Z}$ was arbitrary, it must be the case that $z \notin B_z$ for all $z \in \mathcal{Z}$. This means that $\bigcap_{z \in \mathcal{Z}} B_z = \emptyset$. Then, using the fact that $(\mathcal{Y}, \mathcal{Z}, \ell)$ is a Helly space with Helly number $p$, there exists $p$ balls in $\{B_z\}_{z \in \mathcal{Z}}$ such that their collection-wise intersection is also empty. Accordingly, let $z_1, \ldots, z_p$ be such that $\bigcap_{i=1}^{p} B_{z_i} = \emptyset$. As before, for every $i \in [p]$, let $y_{z_i}$ denote the label on the outgoing edge from $v$ indexed by the Dirac measure $\delta_{z_i}$. By definition, for all $i \in [p]$ and $h_{\delta_{z_i}} \in \mathcal{H}_{\delta_{z_i}}$ we have that

$$h_{\delta_{z_i}}(x) \in \mathrm{B}_\ell(y_{z_i}, r_{z_i}) := \tilde{B}_{z_i}$$

Note that $B_{z_i}$ is the $\gamma'$ expansion of $\tilde{B}_{z_i}$. For each $i \in [p]$, relabel the outgoing edge from $v$ indexed by $\delta_{z_i}$ with the tuple $(y_{z_i}, r_{z_i})$. For each $i \in [p]$, reindex the outgoing edge from $v$ indexed by $\delta_{z_i}$ with $i$. Remove all other edges indexed by measures and their corresponding subtrees. There should now only be $p$ outgoing edges from $v$, each indexed by a number $i \in [p]$ and labeled by a tuple in $\mathcal{Y} \times [0, c]$. Note that $\bigcap_{i=1}^p \mathrm{B}_\ell(y_{z_i}, r_{z_i} + \gamma') = \bigcap_{i=1}^p B_{z_i} = \emptyset$, which matches the second constraint imposed by Definition 16. As for the first constraint on shattering, note that for all $i \in [p]$ and all $h_{\delta_{z_i}} \in \mathcal{H}$, we have that $h_{\delta_{z_i}}(x) \in \tilde{B}_{z_i}$. Thus, the hypothesis that shatters the edges indexed by $\delta_{z_i}$ in the original tree according to Definition 8 also shatters the newly re-indexed and relabeled edges according to Definition 16. Thus, for the root node $v$, both constraints imposed by Definition 16 are met. Recursively repeating the above procedure on the subtrees following the $p$ newly reindexed and relabeled edges results in a $p$-dim tree $\gamma'$-shattered by $\mathcal{H}$ of depth $d$. Thus, $\mathrm{SM}_\gamma(\mathcal{H}) \leq p\text{-dim}_{\gamma'}(\mathcal{H})$ for $\gamma' < \gamma$.