# OpenReview forum: "A Unified Theory of Supervised Online Learnability"
_algorithmiclearningtheory.org/ALT/2025/Conference — ALT 2025_

### Official Review · Reviewer_UcuN · 2024-11-07
**some interesting results that I believe are novel; could benefit from reorganization**

**Rating:** 7
**Confidence:** 3

**Review:**

This paper presents a unified complexity that characterizes learnability in the case of bounded losses. I looked over the main results and believe the proofs are correct; however, I have not yet had a chance to check parts (ii) and (iii) of Theorem 8 nor the proof of Lemma 15.

There recently have been many papers looking at online learnability for various problems. I feel that many similar ideas are being used in these works, both in terms of the algorithms and the proofs, meaning it is not so clear how long this process should go on before we have a work that tries to unify things. This work, which aims to provide a unification (at least for bounded losses) via the Sequential Minimax (SM) dimension, is therefore welcome. I hope it can help refocus future work in online learnability and provide more focus on obtaining the best rates and on obtaining adaptive guarantees (going further than minimax results).

In terms of the novelty of the ideas in the proofs, having seen various works using these types of arguments before, I cannot say I detect a high amount of novelty. However, putting together these arguments is not easy. Also, I believe the proof of the upper bound of Theorem 9 relies on some new tricks (namely, the idea of $\varepsilon_t$-realizability) which are of interest to the learning theory community; I comment on later in my review. In any case, by far the most clear contribution in my eyes is the novel concept of Sequential Minimax dimension. The definition, while long, is intuitive.

Readers of this paper would benefit if the authors could be explicit about the assumptions on the algorithm and the assumptions on the adversary. From Definition 1, I can see that in each round, the algorithm deterministically selects its distribution (conditional on adversary's history of plays thus far). From the notion of regret formalized just before Definition 2, it seems that the adversary is oblivious. Can the authors confirm this? For upper bounds (sufficiency of your new dimension), it seems fine to me that you restrict to this type of deterministic learner and oblivious adversaries. We can apply standard arguments from Prediction, Learning, and Games (Lemma 4.1) to extend to adaptive adversaries. However, for your lower bounds (necessity), are the restrictions you impose on the learning algorithm really OK? Or do you avoid these restrictions for the lower bounds? It is worth at least some discussion in the paper.

In the proof of the upper bound in Theorem 9, the sentence "Let $B$ denote the random variable denoting the randomized prediction of all experts (or their corresponding randomized algorithms)." is not clear. Each expert itself is a randomized algorithm (by virtue of its playing, in each round, a draw from its selected probablity distribution). So, there is one source of randomness. Another source of randomness comes down to the use of MWA. At first, I assumed you meant the randomness due to the random draw (for each expert and for each round) from a probability distribution over the action space $\mathcal{Z}$. However, if that is the case, then there is something wrong with the bound given right after "Then, conditioned on $B$, Theorem 21.11 ... tell us that", since there is no conditioning on the RHS. Here, I am assuming that $E(x_t)$ is a probability distribution, as that is consistent with what MRSOA returns. If you meant that $E(x_t)$ is the random sample drawn from the distribution, then this is *not clear* from the context (especially in light of the LHS of the inequality, which does take a conditional expectation); please write the conditional expectation on the RHS explicitly (assuming the inequality is indeed still correct).

For the proof of Theorem 9's upper bound, could you split this into a two or three main pieces? The proof currently is long and notation-heavy, and those two things together make it difficult to process. To get started, think about how you might write a proof sketch. What are the main pieces for that sketch? You then might consider making a lemma for each piece.

I noted many similarities between Algorithm MRSOA and the algorithm RSOA of Raman et al. (COLT 2023). However, a key difference (which I have not seen before; I did spend some time checking) was the introduction of $\varepsilon_t$, which I view as important to handle the case of regression. In the paper, the authors should comment on the algorithmic predecessors to MRSOA (I guess RSOA is one). Also, in your response to my review, I would like to know if the previous algorithms of the SOA style have used $\varepsilon_t$ variables.


## Regarding Organization and Section 5 of the paper
I felt that Section 5 is a bit fast for a few reasons. First, the meaning of "finite character property" (and why the property is important) was not clear to me. I know this appears in some fundamental works, but without more motivation, it may be hard even for most learning theorists to appreciate this pursuit. Also, I do not know if it would be enough to use the finiteness to argue for computational tractability, due to negative results for computable online learnability for Littlestone classes. Finally, I really think the main interesting technical development (proof of Lemma 15) should be sketched in the main text if this result is worthwhile to include in the paper (or are the ideas involved just not interesting?)

I have a concrete proposal for reorganizing and possibly cutting some material, saving it for more thorough coverage in future work. You could save all of Section 5 for future work if you can resolve the open question regarding necessity (the one stated shortly after Lemma 15). This could be a strong, independent work or maybe could appear in a journal version of your current work. If you cannot resolve the open question, then you could use the current material for a COLT open problem (stating some positive results, and then giving as an open problem being the resolution of necessity for classes with finite $p^*$). The space you save could be used to exhibit how SM dimension can recover previous dimensions, i.e., some of the proofs of parts of Theorem 8. It really is a shame that all of proof of Theorem 8 was left to the appendix. For example, perhaps you might sketch, in the main text, how SM dimension reduces to Sequential Fat-Shattering dimension. If you can demonstrate the ease (not sure if it is easy, actually) with which SM can capture previous dimensions, this will better allow others to leverage your SM results for other online learning problems.



### Additional comments:
1. Page 3, first line: What does it mean to “tightly characterize”; you either characterize learnability or you do not.

2. For the list online classification and measure-shattering dimension papers, you should cite the COLT version of each paper.

3. In Section 5, you write "in Section 3 we showed that the SMdim reduces to... and the p-Set LDim". However, as far as I am aware, you only showed how SMdim to recover Measure Shattering dimension. So, did you actually mean to write SM dimension?

4. You write just before Theorem 8 that the theorem "unifies all major existing results in online supervised learning." I strongly suggest adjusting the writing here. It is true that (i) and (ii) are major; they stand the test of time. However, (iii) and (iv) are rather new and it seems premature to say that these are major results (at least without further explanation). Moreover, one of the most major losses in all of learning theory is the log loss, for which there are results for sequential prediction. Yet, Theorem 8 (nor the entire paper) does not say anything about results for log loss.


### Some minor issues:
- In Definition 4, the only equality (=) should be changed to not equals, which would then match Definition 3.
- In the proof of the upper bound in Theorem 9, first math display: your notation for what I will call an "expert function" also needs to have the round index as a parameter since MRSOA_gamma needs to access not only x_t but also the round index (to have access to [t-1]).

**Paper Award:**

No

---

> ### Author Response · Authors · 2024-11-22
>
> We thank the reviewer for their comments. All minor comments and suggestions will be incorporated in the final version. We address the reviewer's main concerns below.
>
> > From Definition 1, I can see that in each round, the algorithm deterministically selects its distribution (conditional on adversary's history of plays thus far). From the notion of regret formalized just before Definition 2, it seems that the adversary is oblivious. Can the authors confirm this?
>
> Yes, the reviewer is correct in that we only consider oblivious adversaries. We will make this explicit in the final version.
>
> > However, for your lower bounds (necessity), are the restrictions you impose on the learning algorithm really OK? Or do you avoid these restrictions for the lower bounds?
>
> Our lower bounds only apply to learning algorithms that deterministically (as a function of past labeled examples revealed by the adversary) output distributions over the prediction space. In particular, any learning algorithm that does not use the realizations of its past predictions to make future predictions satisfies our definition of a supervised online learning algorithm. For example, learning algorithms that randomly output distributions over the prediction space satisfy our definition as the randomness used to output the distribution can be encoded into the
>
> > In the proof of the upper bound in Theorem 9, the sentence "Let  denote the random variable denoting the randomized prediction of all experts (or their corresponding randomized algorithms)." is not clear...
>
> Each expert uses its own independent source of randomness. In addition, as the reviewer pointed out, MWA also uses its own independent source of randomness.  $B$ is the random variable representing all the randomness used only by the experts to make their predictions. The conditional expectation in the math display above Equation (2) is thus just over the randomness of MWA. Theorem 21.11 of Shalev-Shwartz and Ben-David (2014) is actually pointwise on the expert predictions. In other words, it holds for every possible sequence of expert predictions. So conditioning on $B$ means that we are essentially fixing the predictions of the experts.  We use $E(x_t)$ to denote the actual prediction of the expert at time point $t$ (i.e the random sample), and so it is a random variable (unconditionally). We will make sure to make this explicit in the final version. Accordingly, conditioned on $B$, the RHS is no longer random, since fixing $B$ fixes all the expert predictions in advance. We will make sure to make this explicit in the final version.
>
> > For the proof of Theorem 9's upper bound, could you split this into a two or three main pieces?
>
> Yes, we will give a proof sketch and split the proof of Theorem 9 into multiple pieces for the final version.
>
> > I noted many similarities between Algorithm MRSOA and the algorithm RSOA of Raman et al. (COLT 2023)...
>
> We will make sure to comment the predecessors of MRSOA. These include the standard SOA, Bandit SOA, RSOA, List SOA, RandSOA,  and so on. Indeed, the MRSOA is similar to the RSOA, however, the main difference is the incorporation of $\epsilon_t$-realizability and the loss function when optimizing for the correct distribution to play in each round. In fact, if one picks $\epsilon_t = 0$, then the MRSOA reduces exactly to the RSOA. So, the MRSOA is a generalization. To the best of our knowledge, $\epsilon_t$ variables have not been used by previous algorithms of the SOA-style.
>
> > You write just before Theorem 8 that the theorem "unifies all major existing results in online supervised learning." I strongly suggest adjusting the writing here.
>
> We agree with the reviewer and will tone down this claim to ``unifies several existing results in online supervised learning."
>
> > Regarding Organization and Section 5 of the paper...
>
> We thank the reviewer for their thoughtful comment regarding a reorg. We like the plan proposed by the reviewer and will try our best to implement this in the final version. In particular, we will aim to shorten Section 5 and expand more on the proof of Theorem 8 in the camera-ready version.
>
> > However, as far as I am aware, you only showed how SMdim to recover Measure Shattering dimension. So, did you actually mean to write SM dimension?
>
> Theorem 10 in Raman. et al (2023) proves the equivalence between the MSdim and $p$-Set Ldim for finite Helly spaces. We will make this more explicit in the camera-ready version.

---

> > ### Comment · Reviewer_UcuN · 2024-11-28
> > **lower bound follow-up**
> >
> > Thanks for your responses. I have a follow-up question. You write:
> > > Our lower bounds only apply to learning algorithms that deterministically (as a function of past labeled examples revealed by the adversary) output distributions over the prediction space. In particular, any learning algorithm that does not use the realizations of its past predictions to make future predictions satisfies our definition of a supervised online learning algorithm. For example, learning algorithms that randomly output distributions over the prediction space satisfy our definition as the randomness used to output the distribution can be encoded into the
> >
> > Please carefully think about whether your lower bound proof can be made to work for all learning algorithms (not just some restricted class), as if it cannot be made to do so, then you can no longer claim to characterize learnability. I am less enthusiastic about this work if the characterization of learnability fails to hold. I feel that it must just be an exercise to extend your lower bounds to the class of *all* learning algorithms; specifically, I have in mind those algorithms that use their past realizations to make future predictions; intuitively, using past realizations should not help. I look forward to your response.
> >
> > (Also, note that your response here was truncated ("can be encoded into the..."), but I did get the main idea.)

---

> > ### Author Response · Authors · 2024-11-28
> >
> > We thank the reviewer for their follow-up question. In online learning literature, it is standard convention to define a randomized learner as a sequence of deterministic mappings to probability distributions. For example, see Chapter 4 in ``Prediction, Learning, and Games" by Cesa-Bianchi and Lugosi, Section 1.2.2 of "Online Learning and Online Convex Optimization" by Shalev-Shwartz, and Section 21.2 in "Understanding Machine Learning" by Shalev-Shwartz and Ben-David. Under full-information feedback, online learnability is thus usually only defined with respect to such randomized learners.
> >
> > Nevertheless, the reviewer's intuition is correct in that our lower bound holds even for algorithms which use the realizations of past predictions to output a new measure. The proof is identical except now the adversary computes and uses the ``expected" measure that the learner will play on round $t$ to traverse down the SM tree. We expand on this below.
> >
> > In full generality, a randomized learner is a sequence of maps $f_1, f_2, \dots, f_T$ where $f_1: \mathcal{X} \rightarrow \Pi(\mathcal{Z})$ and $f_t: (\mathcal{X} \times \mathcal{Y})^{t-1} \times \mathcal{Z}^{t-1} \times \mathcal{X} \rightarrow \Pi(\mathcal{Z})$. On round $t$, if the learner's past predictions are $z_1, \dots, z_{t-1}$, then its prediction on round $t$ is $z_t \sim f_t(x_{1:t-1}, y_{1:t-1}, z_{1:t-1}, x_t).$ Now, we can define the "expected" measure on round $t$ as:
> >
> > $$ g_t(x_{1:t-1}, y_{1:t-1}, x_t) := \mathbb{E}\_{z_1 \sim f_1(x_1)}\left[ \mathbb{E}\_{z_2 \sim f_2(x_1, y_1, z_1, x_2)}\left[ \cdots \mathbb{E}\_{z_{t-1} \sim f_{t-1}(x_{1:t-2}, y_{1:t-2}, z_{1:t-2}, x_{t-1})}\left[f_{t}(x_{1:t-1}, y_{1:t-1}, z_{1:t-1}, x_t) \right] \right]\right].$$
> >
> >
> > Note that $g_t(x_{1:t-1}, y_{1:t-1}, x_t)$ is only a function of the data stream $(x_1, y_1), \dots , (x_T, y_T)$ and so it can be computed by the adversary before the game begins. Moreover, because we are considering an oblivious adversary, the expected regret can be written in terms of the "expected" measures, and so our lower bounds applies to this learning algorithm if the adversary uses $g_t$'s to traverse down the SM tree. We will make sure to include a comment about this in the final version.

---

### Official Review · Reviewer_MnWQ · 2024-11-08
**Strong accept; this paper indeed provides a unifying picture of online learnability**

**Rating:** 9
**Confidence:** 4

**Review:**

## Paper Summary
This paper provides a characterization of online learnability with bounded loss
functions via a natural notion of the Sequential Minimax dimension. It directly
generalizes other specific notions of learning dimensions including Littlestone
dimension, fat-shattering dimension, among others. It recovers existing results
up to logarithmic factors.

## Review
This is one of those cases when reviewing a paper does not feel like reviewing a
paper, where one can just enjoy learning from the authors. The authors succeed
on what they claim in the paper title: providing a clear, unifying picture of
the hardness of online learning in various related settings. There are certainly
novel ideas in this paper, but in some sense, the result of the paper is not
extremely surprising. In addition to the main results themselves, I read this
paper as marking the maturity of the area of online learnability, which this
paper brings together and rounds out here. This in itself is a worthy
contribution, and the authors do a wonderful job.

A very nice idea I encountered for the first time here is the idea of
$\epsilon_t$-realizability and the accompanying realizable-to-agnostic
conversion. If this idea came from elsewhere, it would be good to cite that more
obviously. If not, the paper's stated contribution of "identifying the right
notion of realizability and providing a new realizable-to-agnostic conversion"
is even a little understated and could be emphasized more if the author wished
to do so.

The final discussion on the finite character property was also interesting,
although I wasn't quite sure how to place it. Is the main importance for the
computability of the Sequential Minimax dimension? Or, is it the case that
online learning is "easier" when losses take place in a Helly space, for
example?

**Paper Award:**

No

---

> ### Author Response · Authors · 2024-11-22
>
> We are glad the reviewer enjoyed our paper. We address the reviewer's comments and questions below.
>
> > A very nice idea I encountered for the first time here is the idea of $\epsilon$-realizability and the accompanying realizable-to-agnostic conversion ...
>
> We thank the reviewer for finding our notion of $\epsilon$-realizability to be a ``very nice idea." To the best of our knowledge, this idea is new and has not appeared in previous works. We will make sure to emphasize this contribution in the final version.
>
> > Is the main importance for the computability of the Sequential Minimax dimension? Or, is it the case that online learning is "easier" when losses take place in a Helly space, for example?
>
> We thank the reviewer for finding Section 5 to be interesting. The main importance for a dimension to satisfy the finite character property is to have the ability to compute the dimension by probing only finitely many elements from $\mathcal{X}, \mathcal{Y}$ and $\mathcal{F}$ [Ben-David et al. 2019]. This indeed could be useful for computability reasons -  if one assumes that applying $f$ on $x$ requires bounded computation in some reasonable model then checking the property will also require a bounded amount of computation.

---

### Official Review · Reviewer_3Stt · 2024-11-11

**Rating:** 6
**Confidence:** 4

**Review:**

Contributions and Novelty:

This paper provide a unified general framework for analyzing the online learnability of online learning problems. At the core is the introduce of the definition a new combinatorial dimension, named sequence minimax dimension, which is defined with respect to **a specific loss function**. By specifying the loss function and parameter space, the notation reduces to the correct combinatorial dimension for the specic problems, such as Littlestone dimension.

The unified framework is very interesting, which not only covers existing online learnability notations (providing a unified proof) but also has the potential to motivate new ways of characterizing online learnability for other online learning problems. I checked the proof for proving the reduction to other dimensions, which involves bounding those notations from above and below, seems not obvious and non-trival.

Weaknesses/questions:

The authors showed that the proposed dimension notation reduces to existing hardness notations for several problems. While it is nice, it is unclear to me if the new framework can provide any meaningful new results.

Based on the proof of Lemma 19, it seems that the SMdim and MSdim has a very direct connection. I wonder if the authors could add more discussion on this point. Is the definition of SMdim coming from the observation that MSdim can be written in a more general way?

Presentation:

This paper is very well-written and easy to follow. I appritate that the authors explain all of the existing ideas is a very clear way.

Minor comments:

Definition 7 (last line): it seems that $z\sim\mu$ should be $z\sim\mu_t$.

**Paper Award:**

No

---

> ### Author Response · Authors · 2024-11-22
>
> We thank the reviewer for noting that the unified framework ``is very interesting." We address the reviewer's comments and concerns below.
>
> > While it is nice, it is unclear to me if the new framework can provide any meaningful new results.
>
> Our framework provides meaningful new results for several settings. For example, a practical learning setting captured by our model is infinite-dimensional regression between function spaces (e.g. operator learning). Existing combinatorial dimensions and complexity measures for regression do not provide characterizations for this setting because the label space may not be totally bounded. On the other hand, the SMdim provides a characterization of learnability for these settings. Other natural examples without a known characterization include multilabel ranking with the sum loss and multilabel classification with the Hamming loss.  Our framework and results provide characterization for each one of these settings.
>
> > I wonder if the authors could add more discussion on this point. Is the definition of SMdim coming from the observation that MSdim can be written in a more general way?
>
> The reviewer is exactly right in that the SMdim is a generalization of the MSdim to capture general loss functions and to go beyond realizability. We will make sure to comment on this in the final version.

---

### Comment · Area_Chair_cS2c · 2024-11-26
**Connection to previous work**

It will be much appreciated if the authors can describe here the differences between this work and the following recent work that "characterizes online universal learning for bounded losses". It seems that the setup and goals are different but an informative input will help.

Blanchard, Moise. "Universal online learning: An optimistically universal learning rule." Conference on Learning Theory. PMLR, 2022.

---

> ### Author Response · Authors · 2024-11-27
>
> We thank the AC for their question. We highlight several key differences between our work and ``Universal online learning: An optimistically universal learning rule."
>
> - **No restriction on labeling function.** In our setup, there exists a function class $\mathcal{F}$, and the goal of the learner is to drive the expected regret with respect to $\mathcal{F}$ to $0$. In contrast, there is no function class in the work by Blanchard. Instead, the stream is labeled by some unknown measurable function and the goal is to drive the average cumulative loss to $0$ after placing some restrictions on the sequence of instances that can be chosen by the adversary.
>
> - **Restriction on instance sequence.** In our setup, we place no restrictions on the sequence of instances the adversary may reveal to the learner (the restriction is instead placed on how the stream is labeled). In contrast, Blanchard considers a collection of stochastic processes and restrict the adversary to play a sequence of instances sampled according to a process from this set. To quote from Blanchard's paper:
>
> "The first category of works does not restrict the input sequences $X$ but instead the target functions $f^{\star}$ $\dots$ The subject of this paper is of a third category, in which we impose no assumptions on the set of target functions  $f^{\star}$, but instead, restrict the input sequences $X$."
>
>
> - **Same prediction and label space.** In our setup, the prediction space and label space may be different. In contrast, Blanchard only studies the case where the prediction and label space are the same.
>
> We will make sure to include this discussion in the final version.

---

### Meta-Review · Area_Chair_cS2c · 2024-12-09

**Recommendation:** Accept
**Confidence:** 4

**Metareview:**

All reviewers and I agree that the contributions of this paper are solid, interesting to the ML community, and a good fit for ALT 2025.

**Paper Award:**

No